# Sensory perception drives food avoidance through excitatory basal forebrain circuits

Jay M Patel[1,2], Jessica Swanson[3], Kevin Ung[4], Alexander Herman[5], Elizabeth Hanson[3], Joshua Ortiz-Guzman[4], Jennifer Selever[3,6], Qingchun Tong[7], Benjamin R Arenkiel[2,3,4,6]*

[1]Medical Scientist Training Program, Baylor College of Medicine, Houston, United States; [2]Department of Neuroscience, Baylor College of Medicine, Houston, United States; [3]Department of Molecular and Human Genetics, Baylor College of Medicine, Houston, United States; [4]Program in Developmental Biology, Baylor College of Medicine, Houston, United States; [5]Department of Molecular Physiology and Biophysics, Baylor College of Medicine, Houston, United States; [6]Jan and Dan Duncan Neurological Research Institute, Texas Children's Hospital, Houston, United States; [7]Institute of Molecular Medicine, University of Texas Health Science Center, Houston, United States

**Abstract** Appetite is driven by nutritional state, environmental cues, mood, and reward pathways. Environmental cues strongly influence feeding behavior, as they can dramatically induce or diminish the drive to consume food despite homeostatic state. Here, we have uncovered an excitatory neuronal population in the basal forebrain that is activated by food-odor related stimuli, and potently drives hypophagia. Notably, we found that the basal forebrain directly integrates environmental sensory cues to govern feeding behavior, and that basal forebrain signaling, mediated through projections to the lateral hypothalamus, promotes selective avoidance of food and food-related stimuli. Together, these findings reveal a novel role for the excitatory basal forebrain in regulating appetite suppression through food avoidance mechanisms, highlighting a key function for this structure as a potent integrator of sensory information towards governing consummatory behaviors.

DOI: https://doi.org/10.7554/eLife.44548.001

*For correspondence: arenkiel@bcm.edu

## Introduction

Sensory perception of food can induce salivation, increase appetite, and prepare the body for food intake through the release of gastric acid and insulin prior to consumption (*Rogers and Hill, 1989*). Similarly, the perception of rotting or spoiled food can elicit the opposite effect, including diminished appetite and overt food aversion (*Li and Liberles, 2015*). The integration of sensory information to either drive or diminish appetite involves a complex combination of both innate and learned behaviors (*Li and Liberles, 2015*; *Rozin and Fallon, 1987*). Food perception can thus be interpreted as a higher-cortical function, beyond the level of hypothalamic feeding centers, and necessarily requires communication between hypothalamic circuits and extra-hypothalamic regions.

Previous research has shown that hypothalamic feeding-associated neurons are rapidly modulated by sensory perception of food prior to consummatory behaviors (*Betley et al., 2015*; *Chen et al., 2015*). Additionally, nearby neurons within the lateral hypothalamus have also been shown to rapidly drive both appetitive and aversive feeding behaviors (*Jennings et al., 2015*; *Jennings et al., 2013*). How multi-sensory information is relayed to these hypothalamic nuclei to influence feeding or non-consummatory behaviors is not known.

In humans and rodents, olfaction plays a key sensory perceptive role in feeding. Anosmic humans exhibit abnormal food perception and changes in appetite (*Bonfils et al., 2005*). Similarly, olfaction in mice has been shown to impact food intake and metabolism (*Riera et al., 2017*). Additionally, olfactory perception of food itself has been shown to affect neuronal activity in hypothalamic feeding centers (*Chen et al., 2015*). It remains unclear however, how olfactory perception of food is relayed to the hypothalamus to regulate feeding.

A recent non-hypothalamic structure that has been shown to influence feeding is the basal forebrain (*Herman et al., 2016*; *Zhu et al., 2017*). Traditionally identified as a cholinergic center, the basal forebrain has been shown to mediate aspects of wakefulness, attention, learning and plasticity, and sensory modulation (*Golden et al., 2016*; *Hangya et al., 2015*; *Harrison et al., 2016*; *Lin et al., 2015*; *Pinto et al., 2013*; *Ramanathan et al., 2009*; *Xu et al., 2015*). Recent work, has also identified the cholinergic basal forebrain as a robust modulator of appetite (*Herman et al., 2016*; *Zhu et al., 2017*). However, these previous findings did not reveal what aspects of feeding activate basal forebrain neuronal subtypes types, or the direct down-stream cellular connections that mediate feeding behaviors.

Here we identified an excitatory neuronal population within the basal forebrain (BF) marked by vGlut2+ expression (vGlut2$^{BF}$ neurons) (*Dumalska et al., 2008*; *Hur and Zaborszky, 2005*; *Sotty et al., 2003*; *Xu et al., 2015*), that potently influences feeding. Data have shown that vGlut2$^{BF}$ neurons promote wakefulness, and exhibit diverse activation patterns in response to behavioral and training tasks (*Harrison et al., 2016*; *Xu et al., 2015*). In the current study, using a combination of Fos activity measures and in-vivo microendoscopy-imaging methods, we identified activation of vGlut2$^{BF}$ neurons in response to feeding related events and food-odor stimuli. These data, alongside aforementioned work implicating the basal forebrain in appetite control, led us to further examine the role of vGlut2$^{BF}$ neurons in feeding behavior.

Utilizing targeted genetic and viral manipulations of vGlut2$^{BF}$ neurons, we discovered a novel and severe hypophagic phenotype that resulted in dramatic weight loss, starvation, and death, following genetically targeted vGlut2$^{BF}$ neuron activation. Notably, the lethal starvation phenotype could be rescued by nutrient supplement and oral gavage, suggesting that vGlut2$^{BF}$ circuits directly gate behavioral induced hypophagia mechanisms. Further in vivo microendoscopic imaging revealed that both odors associated with rotten food, and aversive odorant signaling through trace-amine-associated receptors in the main olfactory bulb (*Dewan et al., 2013*; *Osada et al., 2018*; *Saraiva et al., 2016*) potently activated vGlut2$^{BF}$ neurons. Cell type-specific synaptic tracing, and channelrhodopsin-assisted circuit mapping, revealed robust innervation of the lateral hypothalamic region by vGlut2$^{BF}$ neurons. Furthermore, optogenetic activation of vGlut2$^{BF}$ cell bodies or their hypothalamic projections was sufficient to induce hypophagia and led to food avoidance upon the olfactory perception of food. Together, these data reveal that vGlut2$^{BF}$ neurons comprise an odor-responsive excitatory neuronal population that projects to the lateral hypothalamus and is capable of reducing feeding behavior by modulating avoidance to food and/or food-related odors.

## Results

### vGlut2$^{BF}$ neurons respond to food odors

The basal forebrain (BF) comprises a heterogeneous population of neurons (*Zaborszky et al., 2015*; *Zaborszky et al., 2012*). In the adult mouse BF, vGlut2 +neurons define a distinct population of excitatory neurons that do not overlap with cholinergic neurons (*Figure 1A*). Re-feeding after an overnight fast activated BF neurons as detected by increased expression of the immediate early gene product Fos (*Figure 1A–B*) (*Herman et al., 2016*). Similar to cholinergic BF neurons, we observed an increase in the percentage of Fos+ VGlut2$^{BF}$ neurons in the fed state compared to periods of fasting (*Figure 1A,C* and *Figure 1—figure supplement 1A–B*). These data indicate that vGlut2$^{BF}$ neurons are a distinct cellular population apart from cholinergic neurons, and are similarly activated by feeding.

To determine what aspect of feeding activates vGlut2$^{BF}$ neurons, we performed in-vivo microendoscopy to image real-time activation patterns of vGlut2$^{BF}$ neurons expressing the genetically encoded calcium indicator GCaMP6m. For this, vGlut2-Cre mice in which Cre recombinase was targeted to the Slc17a6 locus that encodes Vglut2(*Slc17a6* Cre +/-) were stereotaxically injected with

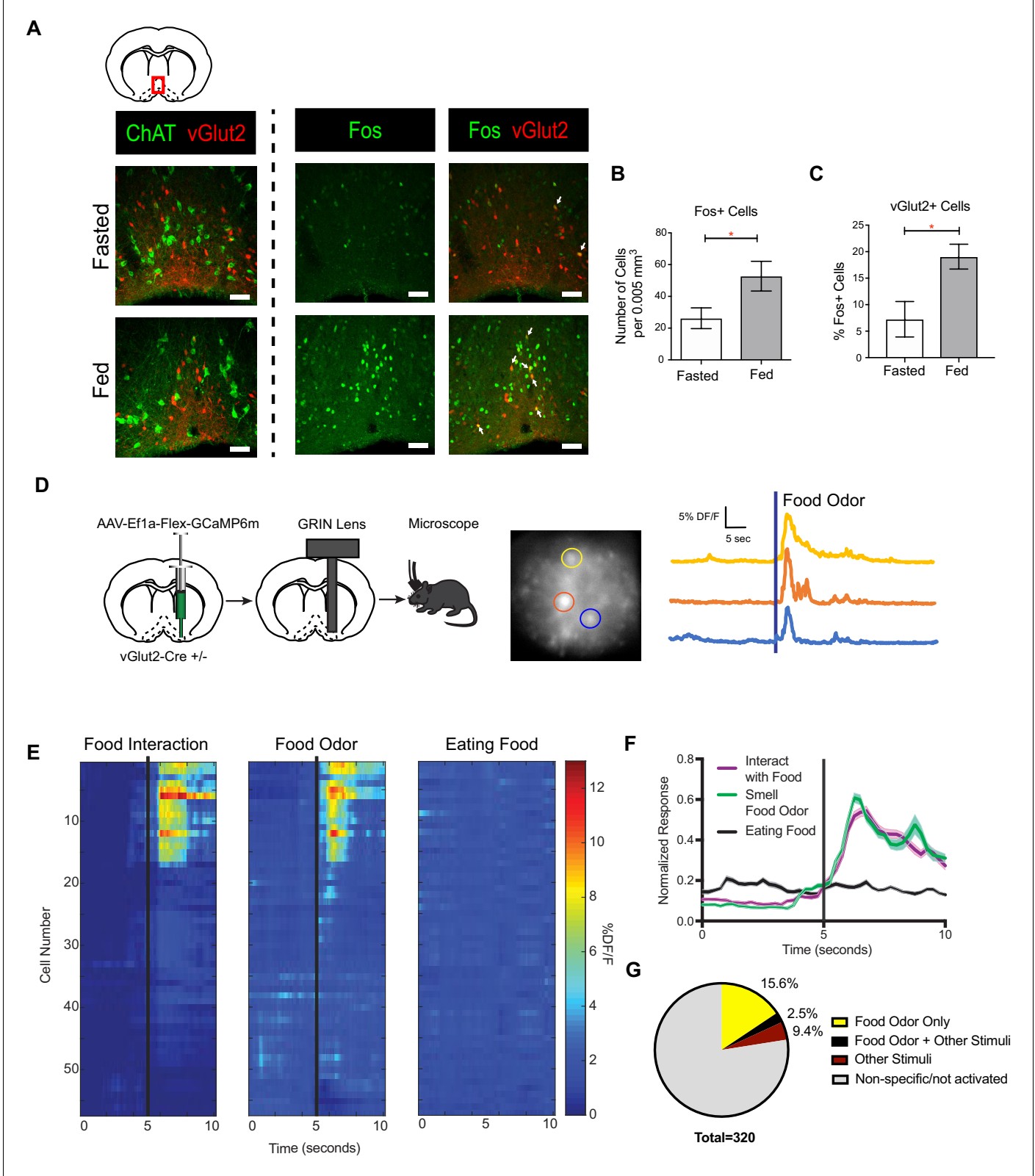

**Figure 1.** vGlut2[BF] neurons are activated by feeding and food associated odors. (A) Coronal diagram showing slice location. Fos and ChAT immunohistochemistry in vGlut2-Cre +/- animals injected with AAV-Flex-mRuby to label vGlut2 +neurons. Scale bar = 50 μm. (B) Quantification of Fos positive cells in diagonal band of Broca (DBB) between fed and fasted animals. *p<0.05, student's t-test, n = 3 animals per group. (C) Percentage of Fos positive DBB VGlut2 +cells in either fasted or fed state. *p<0.05, student's t-test. n = 3 animals per group. scale bar = 50 μm. (D) Schematic of viral

*Figure 1 continued on next page*

*Figure 1 continued*

injection and GRIN lens implantation, example raw image 350 µm x 350 µm, and individual neuronal traces showing changes in GCaMP6m calcium fluorescence in corresponding regions of interest when sniffing food odor. (E) Changes in calcium-mediated fluorescence of 56 neurons from one animal under multiple conditions for one trial. The same cell is in the same row across each condition. Black bars indicate stimulus onset. (F) Combined normalized calcium fluorescence activity from five mice, n = 50 neurons showing neuronal activation to food interaction and/or smelling food odor. (p<0.05). (G) Percentage of total neurons recorded that showed activation to different stimuli. Other Stimuli include physical touch, grooming, and walking.

DOI: https://doi.org/10.7554/eLife.44548.002

The following figure supplement is available for figure 1:

**Figure supplement 1.** Cholinergic[BF] neurons are activated by feeding, and vGlut2[BF] neuron activation patterns.

DOI: https://doi.org/10.7554/eLife.44548.003

---

an adeno-associated virus (AAV) engineered to express Cre-dependent GCaMP6 (AAV-Flex-GCaMP6m) into the basal forebrain (*Atasoy et al., 2008*; *Chen et al., 2013*). We selectively targeted the horizontal limb of the Diagonal Band of Broca (DBB), since previous studies have shown that cell types associated with feeding are connected to the DBB (*Sakurai et al., 2005*; *Wang et al., 2015*). This was followed by implantation of a 500 µm diameter GRIN lens, which allowed recording of calcium dynamics from individual neurons (*Figure 1D* and *Figure 1—figure supplement 1C*). Following recovery, we used miniaturized fluorescence microscopy, combined with high resolution video tracking, to record vGlut2 +neuronal activity in the BF while mice freely moved throughout their environment.

We identified a population of activated vGlut2[BF] neurons when fasted mice approached and interacted with food, prior to consummatory behavior (*Figure 1E* and *Figure 1—figure supplement 1D*). Notably, these same neurons were activated by food odor presentation alone but showed minimal responsivity while eating food (*Figure 1E–F*). Approximately 15.6% of DBB neurons were reactive to food odor alone, with 2.5% being responsive to not only food odors, but also other stimuli that included walking, grooming, or physical stimuli (*Figure 1G* and *Figure 1—figure supplement 1D*). 9.4% of neurons recorded responded to these other stimuli but not food, and the vast majority were unreactive or showed non-specific activation (*Figure 1G* and *Figure 1—figure supplement 1D*). Together, these data identify a population of vGlut2[BF] neurons that are selectively activated by food odors.

## vGlut2[BF] neurons drive hypophagic behavior

Given the responsivity of vGlut2[BF] neurons to food odors, we next sought to selectively manipulate vGlut2[BF] neuronal activity to determine their role in feeding behavior. Towards this, we engineered an AAV that expressed a mouse, codon-optimized version of the sodium bacterial channel, NaChBac (AAV-Flex-eGFP-p2a-mNaChBac), which has previously been used as a genetic approach to upregulate neuronal activity by lowering firing thresholds to increase neuronal tone (*Kelsch et al., 2009*; *Lin et al., 2010*; *Ren et al., 2001*; *Xue et al., 2014*). We stereotaxically injected this virus into the basal forebrains of adult vGlut2-Cre +/- mice, thereby allowing cell type-specific expression in vGlut2[BF] neurons (*Figure 2A*). First, by performing whole cell recordings 7–10 days after viral injection, we measured that vGlut2[BF] neurons that expressed mNaChBac showed increased spontaneous frequencies, action potential half-widths, and action potential voltage thresholds (*Figure 2B–C* and *Figure 2—figure supplement 1B*). Current-injection ramps showed increase area under the activation curve in mNaChBac versus control neurons, indicating neuronal activation was sustained and mNaChBac expressing neurons are capable of increasing the excitability of downstream targets (*Figure 2—figure supplement 1C*). We also performed extracellular single-unit recordings to measure mNaChBac expression effects in vivo. For this, vGlut2-Cre +/- animals were either injected with AAV containing Cre-dependent Channelrhodopsin-2 (ChR2) into the basal forebrain (AAV-Ef1a-Flex-hChr2::EYFP), or a 1:1 mixture of AAV containing Cre-dependent ChR2 and mNaChBac. Using photostimulation as confirmation of electrode placement and cell identification, we recorded spontaneous activity in the basal forebrain, and confirmed increased spontaneous frequency in animals containing mNaChBac (*Figure 2—figure supplement 1D*). Thus, cell type-specific targeting of vGlut2[BF] neurons for expression of mNaChBac allowed for sustained and increased neuronal

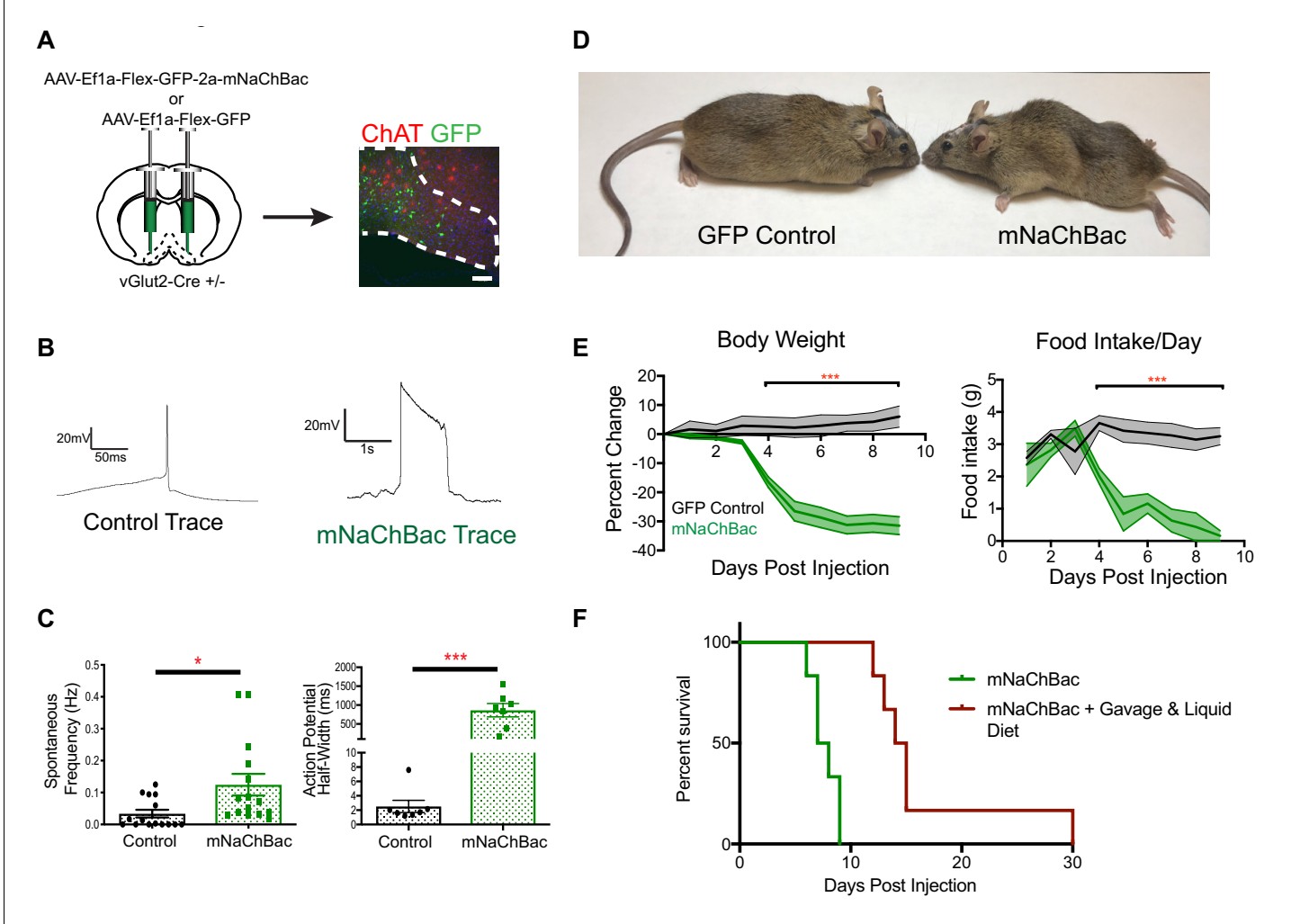

**Figure 2.** Over-activation of vGlut2^BF neurons using mNaChBac results in hypophagia, weight loss, and starvation. (**A**) Coronal brain section showing experimental procedure, viral labeling using eGFP-p2a-mNaChBac, and immunohistochemistry for ChAT in the basal forebrain. Scale bar = 100 μM. (**B**) Example action potential traces from vGlut2^BF neurons 7 days post-injection of mNaChBac virus. (**C**) Spontaneous frequency and action potential half width between control vGlut2^BF and mNaChBac vGlut2^BF neurons 7 days post injection. *p<0.05, ***p<0.001. n = 14 neurons from six animals for frequency and n = 7 neurons from three animals for action potential half-width. (**D**) Representative vGlut2-Cre ± GFP (control) and mNaChBac injected animals 7 days post injection. (**E**) Percentage change in body weight from baseline (average 3 days prior to viral injection) and total daily food intake. ***p<0.001, n = 8 animals 4 male/4 female per group. Data are mean ± SEM. (**F**) Survival curve showing percent animals surviving post injection. n = 6 animals per group. Log-Rank Mantel-Cox test p<0.001. Gehan-Breslow-Wilcoxon test, p<0.01. Median survival 7.5 days vs 14.5 days.

DOI: https://doi.org/10.7554/eLife.44548.004

The following figure supplements are available for figure 2:

**Figure supplement 1.** Electrophysiological properties of vGlut2^BF neurons expressing mNaChBac.

DOI: https://doi.org/10.7554/eLife.44548.005

**Figure supplement 2.** Endocrine hormones are not altered in vGlut2^BF-mNaChBac animals four days after viral injection.

DOI: https://doi.org/10.7554/eLife.44548.006

activation of this population, without the need for exogenous compound delivery (chemogenetics), or fiber-optic implantation for light (optogenetics).

Strikingly, approximately 4–5 days after viral injection, mice began to precipitously lose weight, while concurrently reducing their food intake (*Figure 2D–E*). Experimental animals rapidly lost 30% of their initial body weight as food intake ceased, eventually leading to starvation and death within 9–12 days after viral injection (*Figure 2D–F*). To determine whether the observed weight loss was initially due to either hypophagia, or metabolic dysfunction, we systematically measured a panel of

endocrine hormone levels 4 days following viral delivery, at a time point when animals just began to exhibit weight loss. We found no changes in pituitary hormones, or peripherally circulating thyroid hormones T3 and T4 (*Figure 2—figure supplement 2A–C*). Furthermore, animals did not show altered levels of glucose or insulin, indicating that hypo-insulinemia was not the cause of weight loss (*Figure 2—figure supplement 2D*). Additionally, leptin levels were also similar between experimental mice and controls, indicating that altered leptin signaling did not induce weight loss or decreased food intake (*Figure 2—figure supplement 2D*). In a separate experiment, we also measured ghrelin levels. For this, measurements were taken prior to viral injection under ad libitum and post 12 hr fasting conditions. We then injected half of the experimental mice with mNaChBac virus, while the other half served as GFP-injected controls. In control GFP animals, ghrelin levels were measured once again under ad libitum and 12 hr fast conditions, whereas experimental animals expressing mNaChBac were measured for ghrelin levels 4 days post injection. We detected elevated ghrelin levels in fasted compared to ad libitum fed animals in both pre-injection, and GFP injected animals, indicating viral injection did not alter ghrelin signaling (*Figure 2—figure supplement 2E*). Interestingly, mNaChBac injected animals exhibited elevated levels of ghrelin comparable to the fasted state, indicating ghrelin was indeed being released due to diminished food intake. However, elevated ghrelin signaling was unable to induce feeding, as mice continued to exhibit self-starvation (*Figure 2—figure supplement 2E*). Together, these data suggest that the observed rapid weight loss in mice with increased vGlut2$^{BF}$ neuron activity was primarily due to diminished food intake, and not an altered metabolism. This was further supported by transitioning a group of mice to a liquid diet prior to viral injection, which allowed us to orally gavage a subset of experimental mice and test whether food supplementation could rescue starvation, and/or prolong life in the hypophagic mice. Indeed, we found that food supplementation via oral gavage 2x/day ameliorated the starvation phenotype (*Figure 2F*). Together, these data further substantiate that over-activity of vGlut2$^{BF}$ neurons leads to hypophagia-induced weight loss.

To query the necessity of vGlut2$^{BF}$ neurons in controlling food intake, we next implemented chemogenetic inhibition. Towards this, we stereotaxically injected AAV engineered to express the inhibitory hM4Di DREADD receptor in a Cre-dependent manner (*Armbruster et al., 2007*) (AAV-Ef1a-Flex-hM4Di-mCherry) into the basal forebrains of adult vGlut2-Cre ± mice (*Figure 3A*). Three weeks post injection, we measured food intake of mice eating ad libitum, or following a 24 hr fast. Mice were then IP injected with either CNO or saline as a control, and again monitored for levels of food intake under normal feeding and fasted conditions. In both scenarios, mice treated with CNO displayed increased food intake compared to saline-treated controls (*Figure 3B–C*). Collectively, these data reveal a novel role for vGlut2$^{BF}$ in regulating food intake and identified this neuronal population and their associated circuits as potent drivers of appetite suppression.

## vGlut2$^{BF}$ neuron induced hypophagia is resistant to leptin deficiency hyperphagia

We next sought to determine whether chronic activation of vGlut2$^{BF}$ neurons could ameliorate excessive weight gain in previously described hyperphagic mouse models. For this, we implemented leptin-deficient mice (*Lep$^{ob/ob}$*) which show hyperphagia and become obese through increased food intake and subsequent altered metabolism (*Drel et al., 2006*; *Ingalls et al., 1950*; *Lindström, 2007*). First, we generated vGlut2-Cre+/-;Lep$^{ob/ob}$ mice, then virally targeted vGlut2$^{BF}$ neurons in obese animals (mean weight 51.56g ± 8.22 standard deviation) with the conditional mNaChBac AAV for cell type-specific activation of these neurons. Notably, we found that mice with overactive vGlut2$^{BF}$ neurons via mNaChBac expression suppressed leptin deficiency-induced hyperphagia, and dramatically reduced their food intake with subsequent weight loss (*Figure 4A–B*). Together, these experiments reveal that food avoidance pathways driven by vGlut2$^{BF}$ neurons can over-ride robust intrinsic signaling mechanisms that govern feeding.

## vGlut2$^{BF}$ neurons are activated by naturally aversive odorants

Given that food odors are often considered to be positive stimuli, it was intriguing that activation of food-responsive vGlut2$^{BF}$ neurons resulted in severe hypophagia and starvation. To further investigate this, we performed in vivo calcium imaging as above; however, this time in awake head-fixed animals. We first injected vGlut2-Cre+/+ animals with AAV-Flex-GCaMP6m into the BF, and two

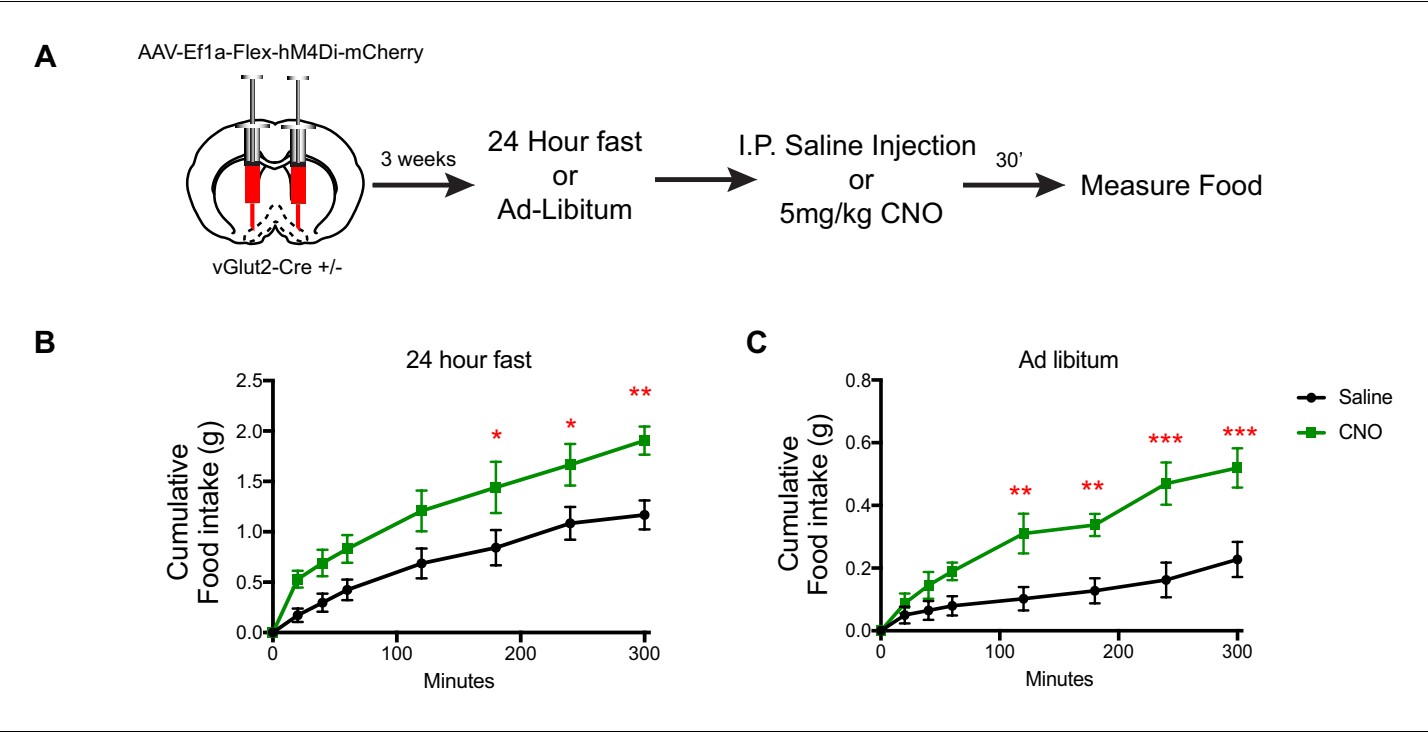

**Figure 3.** DREADD inhibition of vGlut2$^{BF}$ neurons increases food-intake. (**A**) Schematic showing experimental paradigm. (**B**) Measurements of cumulative food intake in experimental and control mice following a 24 hr fast. (**C**) Measurements of cumulative food intake in experimental and control mice following ad-libitum over-night food access. Statistical measures include 2-way ANOVA multiple comparisons. Data are mean ± SEM. N = 5 animals per group. *p<0.05, **p<0.01, ***p<0.001.
DOI: https://doi.org/10.7554/eLife.44548.007

weeks later animals were placed on a head-fixed imaging apparatus that allowed for animals to move on a freely spinning wheel. Acutely, GRIN lenses were lowered over the BF, and cells were imaged using the same DORIC microendoscope as above. This setup allowed for precise odor delivery, and incorporation of multiple trials per animal. We presented mice with mineral oil as a control,

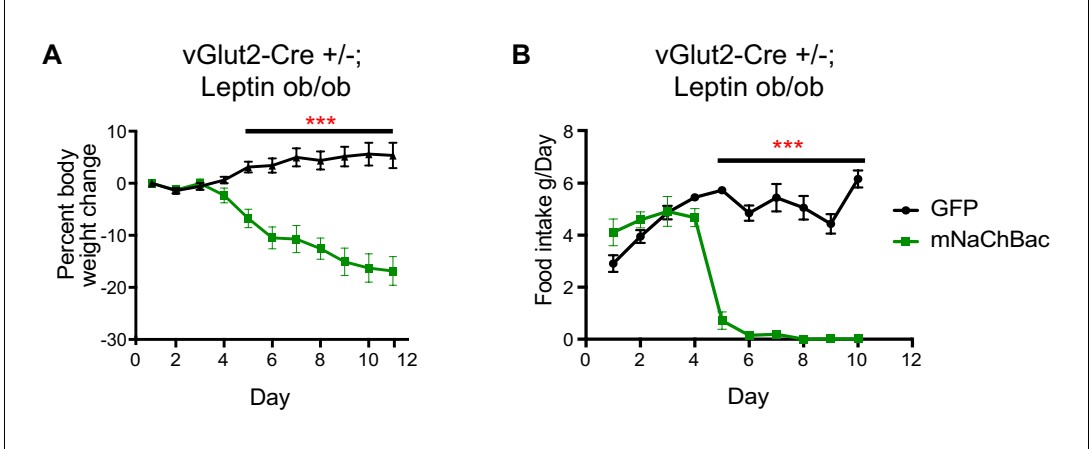

**Figure 4.** Chronic vGlut2$^{BF}$ activation causes hypophagia in *Lep$^{ob/ob}$* mice. (**A**) Percent weight change from baseline in vGlut2-Cre+/-; *Lep$^{ob/ob}$* mice injected with either AAV-Flex-GFP (control) or AAV-Flex-eGFP-p2a-mNaChBac into the basal forebrain. N = 5 animals per group (three males/2 female) (**B**) Food intake in the same animals from (**A**) N = 5 animals per group. ***p<0.001, 2-way RM ANOVA with Sidak's post-hoc. Data are mean ± SEM.
DOI: https://doi.org/10.7554/eLife.44548.008

food odor (chow dissolved in mineral oil), and naturally aversive odorants, including butyric acid - which is normally associated with rotten/spoiled food, and 2-methylbutylamine - a known trace-amine associated receptor ligand (*Dewan et al., 2013*; *Osada et al., 2018*; *Saraiva et al., 2016*). We found that neurons were fairly unresponsive between odor presentations, and different neurons strongly responded to different groups of odors (*Figure 5A*). We also noted that food odors activated vGlut2$^{BF}$ neurons similar to what we observed in freely behaving animals (*Figure 1*). Interestingly, these same neurons also responded strongly to the two aversive odorants (*Figure 5A,B,C*), and overall a greater percentage of vGlut2$^{BF}$ neurons were responsive to the aversive odorants compared to food and/or control stimuli (*Figure 5D*). Together, these data indicate that vGlut2$^{BF}$ neurons represent a heterogenous population of neurons capable of responding to diverse odors with different valences, and that a greater number of the cells more robustly respond to aversive odorants.

## vGlut2$^{BF}$ neurons receive cholinergic input through nicotinic receptors

Cholinergic signaling from the basal forebrain has previously been implicated in appetite suppression (*Herman et al., 2016*). Activating cholinergic neurons in the basal forebrain causes cessation of feeding, while selectively activating their projections to the hypothalamus reduces feeding, but to a lesser extent (*Herman et al., 2016*). Further, nicotinic acetylcholine receptor activation in vivo is a known appetite suppressant (*Mineur et al., 2011*). Since vGlut2$^{BF}$ neurons are a strong driver of appetite cessation, we hypothesized that they may represent an additional downstream target of cholinergic appetite suppression. To examine the synaptic connectivity from cholinergic neurons onto local glutamatergic projection neurons within the basal forebrain, we combined cell type-specific optogenetic activation of cholinergic neurons, with conditional viral labeling to visually target and record from vGlut2$^{BF}$ neurons. Towards this, we generated *Chat-ChR2+/-; vGlut2-Cre +/-* mice that expressed channelrhodopsin in cholinergic neurons, which allowed us to virally target vGlut2$^{BF}$ neurons with a conditional virus engineered to express the red fluorescent protein mRuby2 (AAV-Flex-mRuby2) (*Figure 6A*). When performing whole-cell recordings from red labeled vGlut2$^{BF}$ neurons while simultaneously photostimulating cholinergic inputs, we found that approximately 10% of patched vGlut2$^{BF}$ neurons received fast, monosynaptic, nicotinic currents (six nicotinic responses from 52 recordings) (*Figure 6B–C*). In the presence of TTX and 4-AP, photo-evoked currents were blocked by the nAChR-specific antagonist mecamylamine, whereas the mAChR-specific antagonist atropine did not affect photo-evoked post-synaptic responses (*Figure 6B*). Together, these data suggest that a subset of cholinergic basal forebrain neurons are capable of acting via fast nicotinic signaling onto vGlut2$^{BF}$ neurons, to potentially influence appetite suppression.

## Basal forebrain excitatory neurons project onto vGlut2 +lateral hypothalamic neurons

To determine whether vGlut2$^{BF}$ neurons functionally connect with known feeding centers in the brain to mediate appetite suppression, we performed targeted anterograde synaptic labeling using an AAV engineered to express Cre-dependent GFP-fused synaptophysin (AAV-Ef1a-Flex-Synaptophysin::eGFP), which localizes to presynaptic terminals. For this, we stereotaxically injected AAV-Ef1a-Flex-Synaptophysin::eGFP into the basal forebrains of vGlut2 Cre +/-mice, followed by imaging vGlut2$^{BF}$ neuron projection fields (*Figure 7A*). We observed dense projections to the lateral hypothalamus, with no GFP fluorescence in nearby structures such as the paraventricular nucleus or fornix (*Figure 7B*), as previously described (*Do et al., 2016*). Other prominent regions where we identified terminal fields included the piriform cortex, olfactory bulb, lateral habenula, and ventral tegmental area (*Figure 7—figure supplement 1A–C*). Relative intensities of these regions highlighted the lateral hypothalamus (LH) as a robustly innervated target of vGlut2$^{BF}$ neurons (*Figure 7—figure supplement 1B*), and previous data have implicated the LH as a potent modulator of both feeding and non-consummatory behaviors (*Jennings et al., 2015*; *Jennings et al., 2013*; *Stamatakis et al., 2016*).

Based on these anatomical data, we sought to determine if functional connectivity between vGlut2$^{BF}$ neurons and vGlut2$^{LH}$ neurons could serve as a circuit to control appetite (*Jennings et al., 2013*; *Stamatakis et al., 2016*). To test whether vGlut2$^{BF}$ neurons synapse onto vGlut2$^{LH}$ neurons, we dual-injected vGlut2 Cre +/- mice with AAV containing Cre-dependent ChR2 into the basal

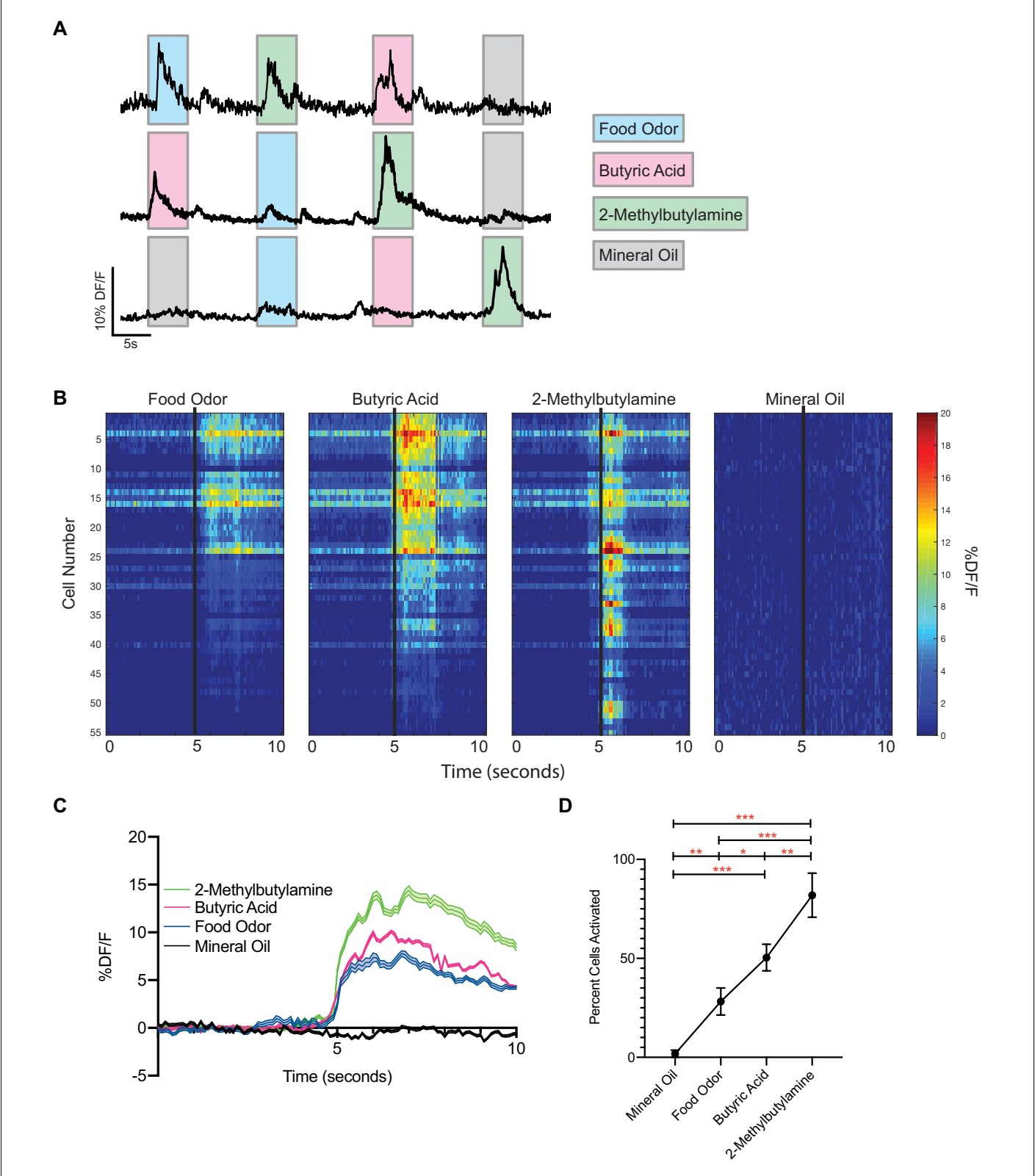

**Figure 5.** vGlut2$^{BF}$ neurons are activated by aversive odorants. (**A**) Example calcium traces from vGlut2$^{BF}$ neurons to different odors presented for 5 s each. (**B**) Representative heat map of all cells recorded from one animal showing % DF/F responsivity to different odors over a 10 s window. (**C**) Combined calcium fluorescence activity from n = 5 mice showing neuronal activation to different odors (p<0.05 2-way ANOVA with post-hoc Tukey's) *Figure 5 continued on next page*

*Figure 5 continued*

(64 mineral oil cells, 142 food odor cells, 168 butyric acid cells, 310 2-methylbutylamine cells). (D) Percent cells activated in each animal to different odors n = 5 mice. *p<0.05, **p,0.01, ***p<0.001, RM one-way ANOVA with Tukey's post-hoc analysis.

DOI: https://doi.org/10.7554/eLife.44548.009

forebrain (AAV-Ef1a-Flex-hChR2::EYFP), and a Cre-dependent red fluorescent reporter into the LH (AAV-Ef1a-Flex-mRuby2) (*Figure 7C*). Using slice electrophysiology, we made acute coronal slices that contained the lateral hypothalamus and patched onto red-labeled vGlut2$^{LH}$ neurons. Through optogenetic mapping, and in the presence of TTX and 4-AP, we found that stimulation of vGlut2$^{BF}$ nerve terminals in the LH resulted in monosynaptic glutamatergic neurotransmission onto vGlut2$^{LH}$ neurons in approximately 88% of cells recorded (8 out of 9 cells recorded) (*Figure 7D–E*). These data directly link vGlut2$^{BF}$ neurons to previously reported vGlut2$^{LH}$ neurons as a possible upstream circuit node to control appetite suppression.

## Optogenetic activation of vGlut2$^{BF}$ neurons or their lateral hypothalamic projections decreased food intake

To further interrogate the functional connectivity between VGlut2$^{BF}$ neurons and the lateral hypothalamus, and to determine if this circuit influences feeding behavior, we next stereotaxically injected an AAV expressing Cre-dependent ChR2 AAV (AAV-Ef1a-Flex-hChR2) into the basal

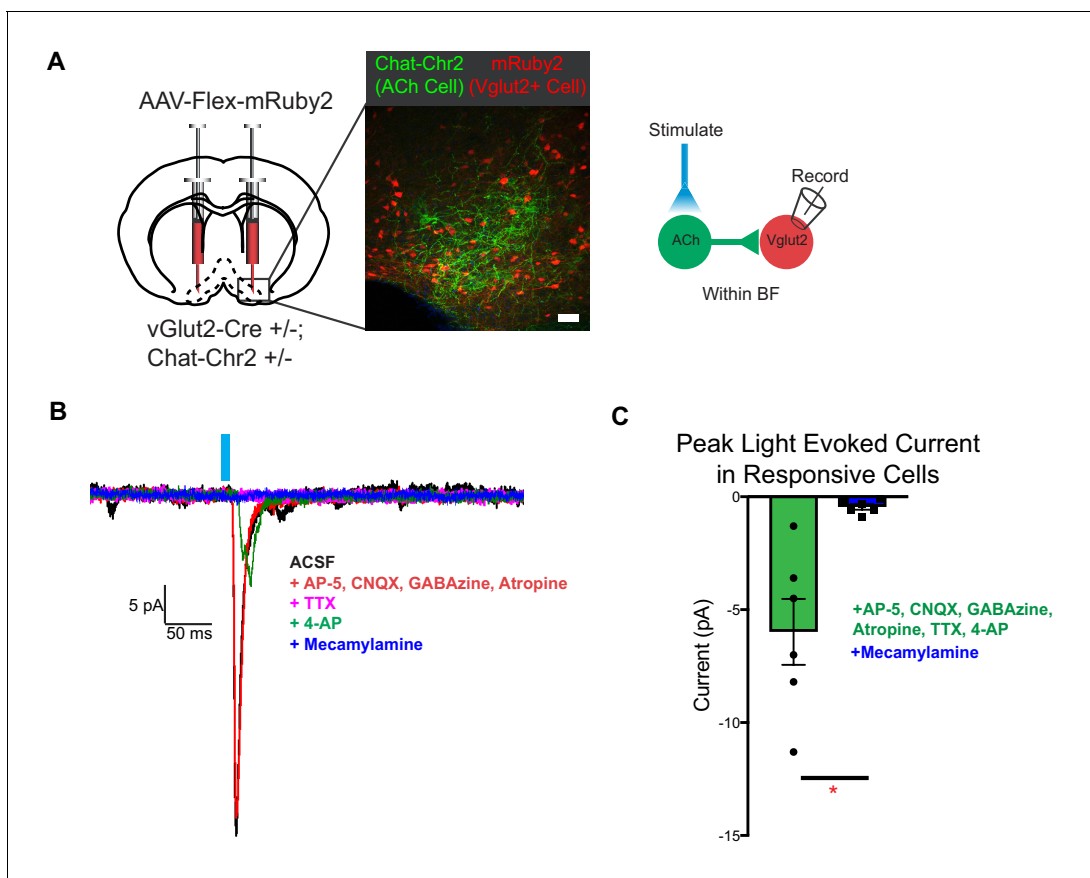

**Figure 6.** vGlut2$^{BF}$ neurons receive fast mono-synaptic nicotinic current. (A) Schematic showing experimental paradigm with confocal picture of basal forebrain labeling cholinergic channelrhodopsin neurons (green) and vGlut2$^{BF}$ neurons (red). Scale bar = 100 μm. (B) Example trace from a vGlut2$^{BF}$ neuron with cholinergic neuron photostimulation. (C) Average evoked monosynaptic nicotinic current onto vGlut2$^{BF}$ neurons. N = 6 neurons out of 52 patched neurons from three animals. *p<0.05, student's t-test.

DOI: https://doi.org/10.7554/eLife.44548.010

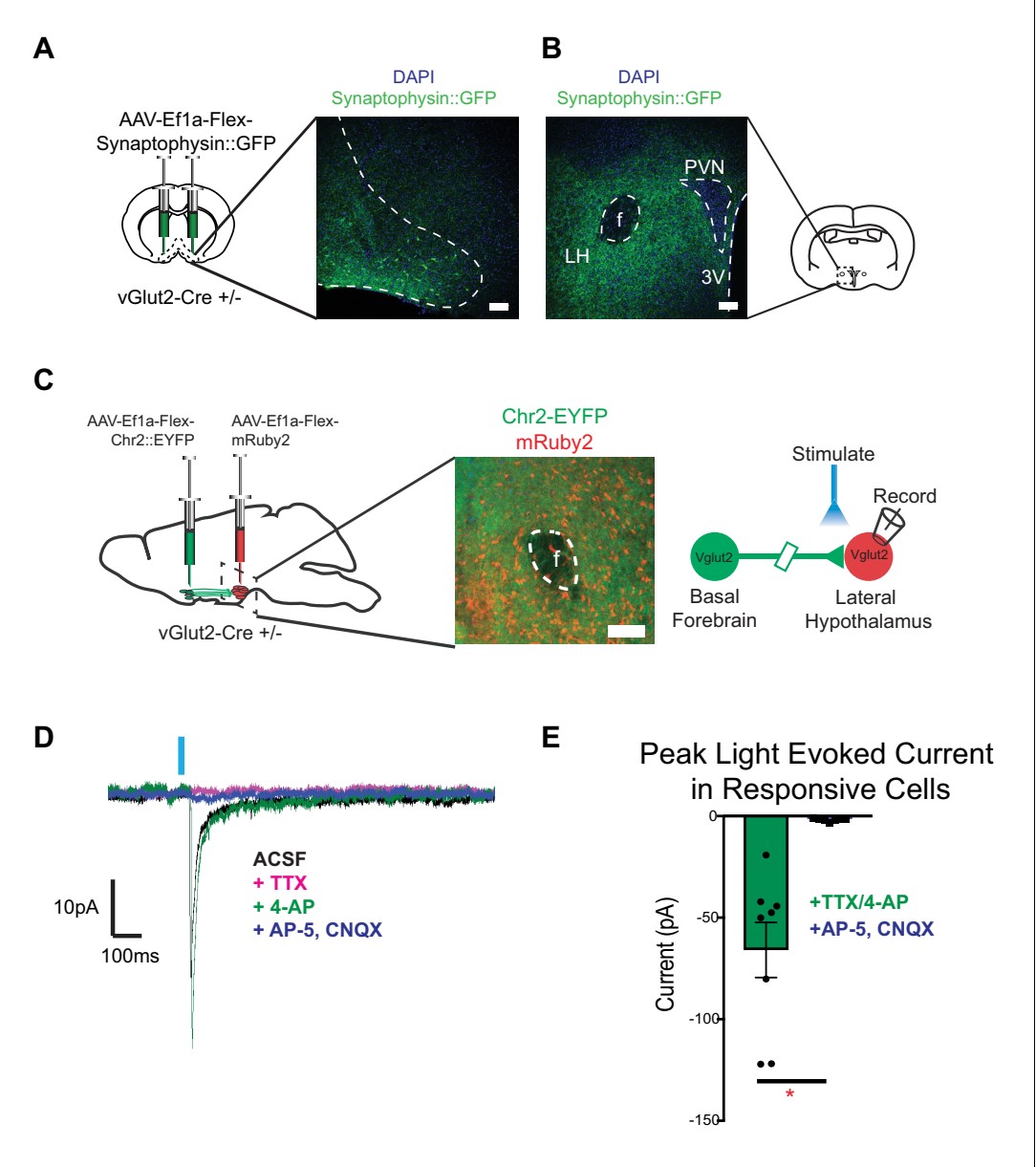

**Figure 7.** vGlut2[BF] neurons project to lateral hypothalamus and make monosynaptic connections onto vGlut2[LH] neurons. (**A**) Coronal diagram showing viral injection site, and immunofluorescence image showing labeled cell-bodies in the basal forebrain. Scale bar = 100 μm (**B**) Coronal slice through hypothalamus showing synaptophysin:: GFP nerve terminals. PVN = paraventricular nucleus, 3V = 3rd ventricle, f = fornix. LH = Lateral hypothalamic area. Scale bar = 100 μm. (**C**) Schematic for dual injection electrophysiology experiment. 300 μm thick acute slice showing green channelrhodopsin containing terminals from vGlut2[BF] neurons and red vGlut2[LH] cell bodies. Scale bar = 100 μm. (**D**) Example traces from a vGlut2[LH] neuron following photostimulation of vGlut2[BF] neuron terminals. (**E**) Average evoked monosynaptic glutamatergic current onto vGlut2[LH] neurons from vGlut2[BF] neurons. N = 8 neurons out of 9 patched neurons from three animals. *p<0.05, student's t-test.

DOI: https://doi.org/10.7554/eLife.44548.011

The following figure supplement is available for figure 7:

**Figure supplement 1.** vGlut2[BF] neurons project throughout the brain.

DOI: https://doi.org/10.7554/eLife.44548.012

forebrains of VGlut2 Cre ±mice (*Figure 8A*), followed by surgical implantation of bilateral fiber optics over both the basal forebrain and lateral hypothalamus, thereby allowing for differential stimulation of vGlut2$^{BF}$ cell-bodies or their LH projections (*Figure 8A*). Whole-cell patch recording on vGlut2$^{BF}$ cell-bodies demonstrated high-fidelity light-evoked action potentials at 5 Hz (*Figure 8B*) and represented a similar frequency utilized by other groups when photo-stimulating vGlut2 +neurons (*Jennings et al., 2013*). Following a 24 hr fast, implanted mice were presented with food, while concurrently photostimulated at 5 Hz either over the BF or LH. Compared to unstimulated control conditions, both BF and LH photo-stimulation resulted in approximately fifty percent reduction of food intake (*Figure 8C*). Increasing BF photo-stimulation frequencies led to a further reduction in food intake (*Figure 8D*), whereas photostimulation did not change the amount of food consumed by control GFP injected animals (*Figure 8C*). Together, results from these optogentic

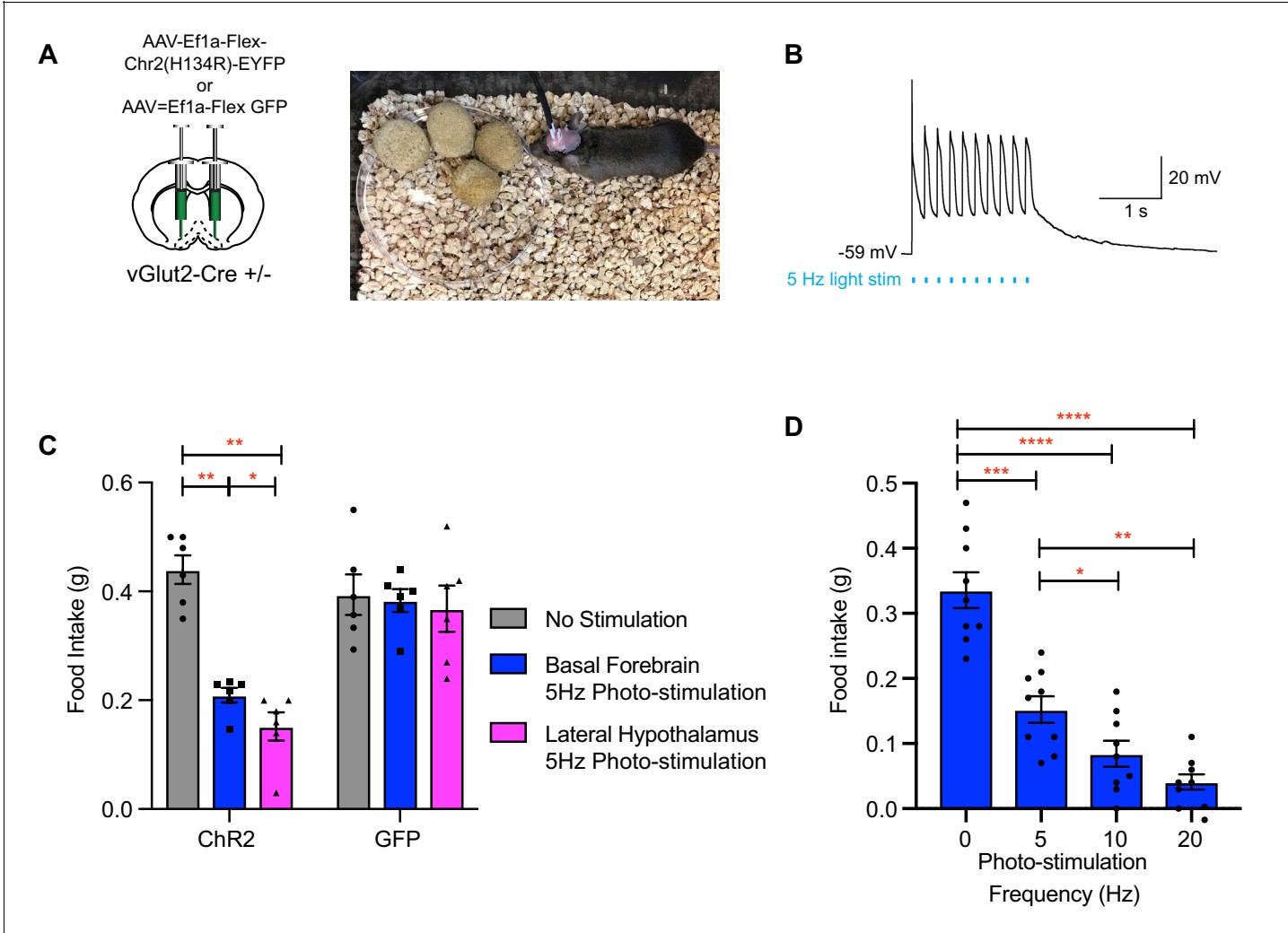

**Figure 8.** Photo-stimulation of vGlut2$^{BF}$ neuron cell bodies and vGlut2$^{BF}$ lateral hypothalamic projections decreases food-intake. (A) Schematic showing viral injection into the basal forebrain (left), and a picture of a mouse with bilateral fiberoptic implants over both the basal forebrain and lateral hypothalamus (right). (B) Whole cell recordings from a channelrhodopsin expressing vGlut2$^{BF}$ neuron showing high fidelity activation with 5 Hz photo-stimulation. (C) Food intake of vGlut2$^{BF}$ channelrhodopsin or GFP expressing mice during 30 min of photo-stimulation. *p<0.05, **p<0.01. Two-way RM ANOVA, post-hoc Tukey's multiple comparison. N = 6 animals, three males/3 females. (D) Food intake of fasted vGlut2$^{BF}$ channelrhodopsin expressing animals implanted only over the basal forebrain and subsequently photo-stimulated only over the basal forebrain at various photo-stimulation frequencies. *p<0.05, **p,0.01, ***p<0.001, ****p,0.0001. RM One-way ANOVA, post-hoc Tukey's multiple comparison, N = 9 animals, five males/4 females.

DOI: https://doi.org/10.7554/eLife.44548.013

experiments corroborate findings from chronic activation mNaChBac, and further implicate the lateral hypothalamic projections of VGlut2$^{BF}$ neurons in mediating appetite suppression.

## Acute activation of vGlut2$^{BF}$ neurons or their LH projections results in food avoidance

The temporal precision of optogenetics in previous experiments allowed us to monitor animal behavior during photostimulation. Interestingly, while performing these manipulations, we observed that experimental ChR2 animals tended to avoid food pellets during periods of photostimulation, while control GFP animals would spend most of their time eating during photostimulation, as would be expected following a 24 hr fasting period. This led us to question whether the dramatic hypophagic phenotype that we observed while manipulating vGlut2$^{BF}$ neurons, may in fact be attributed to food avoidance. To directly investigate this, we placed food pellets in 3 corners of a square arena, while leaving one corner devoid of food. Whereas fasted control animals spent most of the monitored time near and/or eating food pellets, and very little time in the non-food corner, experimental animals with optogenetic stimulation of vGlut2$^{BF}$ neurons showed a strong avoidance to areas of the arena that harbored food (*Figure 9A–B*). We photo-stimulated animals at 5 Hz since we observed a phenotype at this frequency, and it was comparable to the in vivo frequency we measured in mNaChBac animals using extra-cellular single unit recordings (*Figure 2—figure supplement 1D*). In fact, prolonged photo-stimulation of either the basal forebrain, or the terminal projection field within the lateral hypothalamus, led to a preference for the non-food corner (*Figure 9A–B*). Notably, fasted GFP injected animals showed no preference to the non-food corner and spent the majority of their time interacting with food (*Figure 9A–B*). Collectively, these data suggest that the induced hypophagia elicited by targeted activation of vGlut2$^{BF}$ neurons or their projections to LH, may be attributed to a circuit-specific food avoidance.

## Acute activation of vGlut2$^{BF}$ neurons and their LH projections drives avoidance of food odors

Given that we identified that both food and aversive odors activated vGlut2$^{BF}$ neurons, and that targeted manipulation of their activity elicited hypophagia and food avoidance, we next postulated that olfactory perception of food alone might also induce avoidance. To test this, we allowed mice expressing ChR2 in vGlut2$^{BF}$ neurons with bilateral implants over basal forebrain cell bodies and lateral hypothalamic projections to form a passive association of food with the neutral odor R-limonene by spiking mouse chow for two weeks (*Figure 8A*). Following the odor association period, we then photostimulated either vGlut2$^{BF}$ neuron cell bodies or their LH projections, while cotton swabs containing R-limonene were placed in 3 corners of the mouse arena (*Figure 10A–B*). Compared to non-stimulated conditions, optogenetically manipulated mice preferred the empty, non-odor corner. Importantly, swapping odorants to an unassociated odorant S-limonene, the enantiomer to R-limonene that mice can differentiate, did not lead to any detectable odor avoidance (*Figure 10A–B*). Together, these data indicate that photostimulation alone does not cause mice to prefer an empty corner (S-limonene experiment), and that photostimulation of vGlut2$^{BF}$ neurons or their projections to the LH during odor perception of food (R-limonene) can lead to a potent food-odor avoidance behavior.

## Discussion

Recent work has implicated discrete cell types in the basal forebrain as potent regulators of appetite and feeding. Cholinergic basal forebrain neurons have been shown to suppress appetite, while inhibitory VGAT-positive and somatostatin positive neurons promote general and high caloric food intake, respectively (*Herman et al., 2016*; *Zhu et al., 2017*). Here, we uncovered a previously unknown population of basal forebrain vGlut2 +excitatory neurons as extremely potent suppressors of appetite, some of which receive input from neighboring cholinergic neurons. We also found that a subset of these excitatory neurons responded to food-related odors, and an even larger portion of them responded to odors associated with aversive and/or spoiled food odorants. Therefore, it is plausible that by over-activating these neurons, we induced an overriding hypophagic state, where a typical rewarding food pellet became unappetizing. What remains unclear, is whether the valence assigned to consummatory odors versus aversive odors occurs at the level of the basal forebrain.

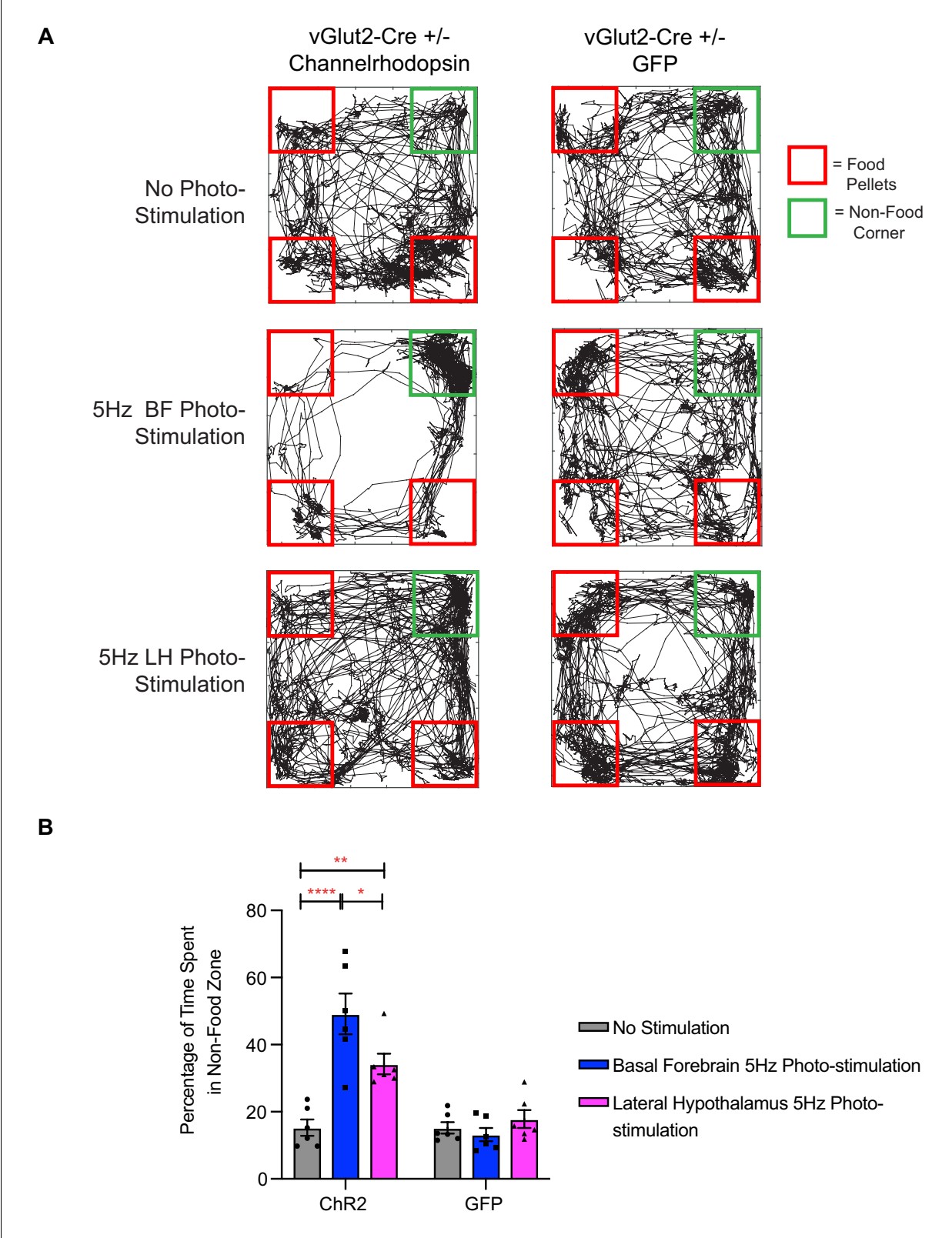

**Figure 9.** Photo-stimulation of vGlut2[BF] neuron cell bodies and vGlut2[BF] lateral hypothalamic projections induces food-avoidance behavior. (A) Example locomotor traces of the same channelrhodopsin and control GFP injected vGlut2-Cre animals under three different conditions for 30 min. (B) Quantification of percentage time spent by channelrhodopsin-expressing (left) or control mice (right) in non-food corner during 30 min of photo-

*Figure 9 continued on next page*

*Figure 9 continued*

stimulation. *p<0.05. **p<0.01, ****p<0.0001. Two-way RM ANOVA, post-hoc Tukey's multiple comparison test, n = 6 animals. three males/3 females. Data are mean ± SEM.

DOI: https://doi.org/10.7554/eLife.44548.014

Notably, given that activating vGlut2[BF] neurons results in hypophagia, it is unlikely that valence assignment occurs downstream of the circuit. Further investigation into these sensory response features will be required to more fully understand vGlut2[BF] neuron inputs from higher cortical regions, as well as diverse outputs to given stimulation paradigms. These data further add to the growing evidence of the importance of the basal forebrain as an integrator of multiple sensory stimuli. Additionally, non-cholinergic cells in the basal forebrain have been found to predict behavioral responses associated with attention, and their activity can be modulated by multiple sensory stimuli in a time scale of sub-seconds to seconds (*Hangya et al., 2015*; *Harrison et al., 2016*). Our imaging data support these claims as vGlut2[BF] neurons were responsive to cues harboring both positive or negative valences, which would be important responses in an attentive state. Together, these findings reveal the robust interconnected dynamics of basal forebrain signaling in governing food intake, appetite, and downstream metabolic change.

Nicotine is a potent appetite suppressor (*Stojakovic et al., 2017*). Recent work has identified key roles for nicotinic signaling in suppressing appetite through POMC neuron activation within the arcuate nucleus of the *hypothalamus* (*Mineur et al., 2011*; *Picciotto et al., 2012*). Our electrophysiology results indicate that vGlut2[BF] neurons functionally express acetylcholine nicotinic receptors, and thus

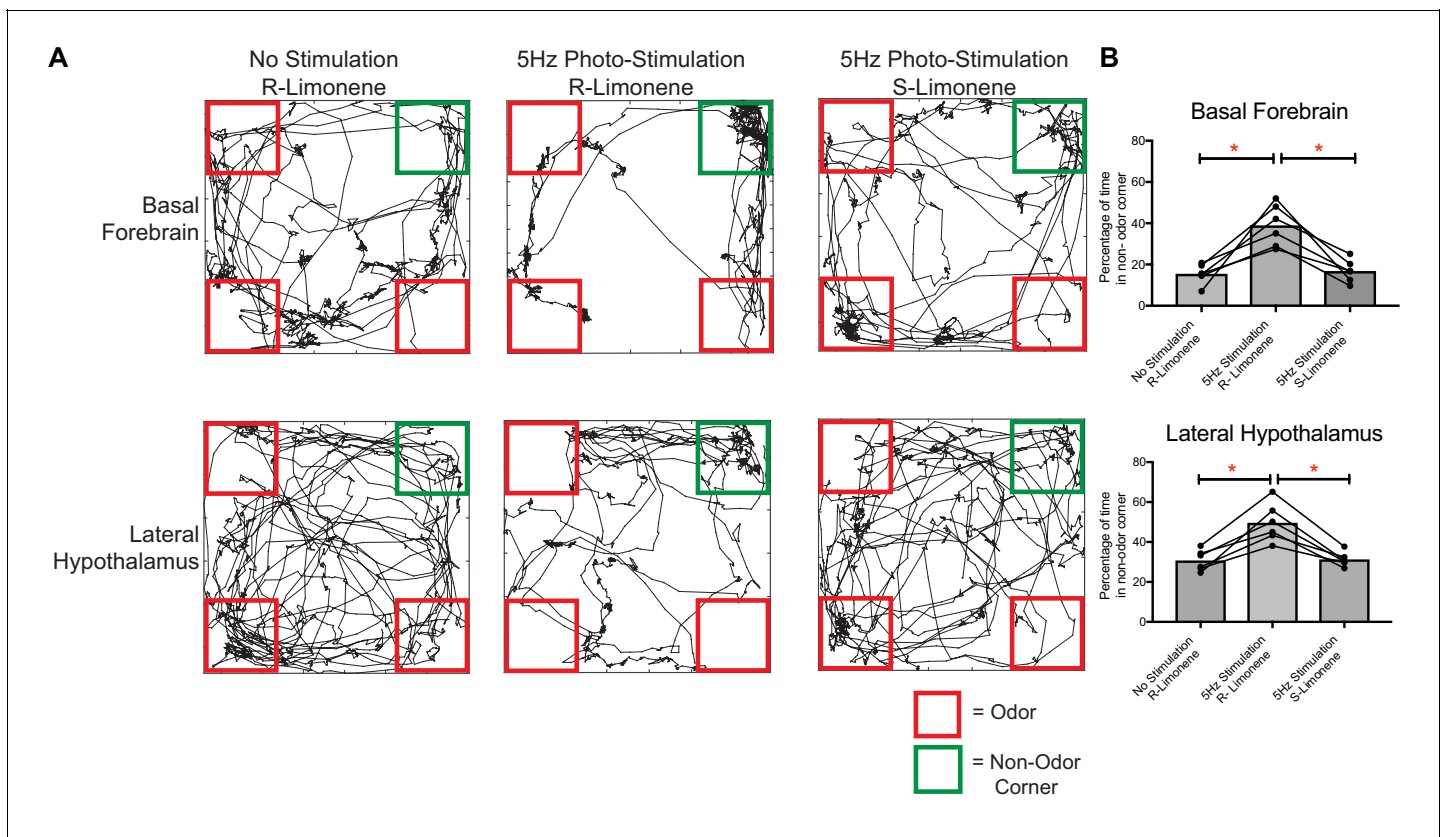

**Figure 10.** Food-associated odor induces avoidance behavior in vGlut2[BF] neuron and vGlut2[BF] LH projection photostimulated animals. (**A**) Locomotor trace of the same channelrhodopsin mouse photo-stimulated either over the basal forebrain or lateral hypothalamus under three different conditions. R-Limonene is passively associated with food, whereas S-Limonene represents the non-associated control. (**B**) Quantification of percentage of time spent in non-odor corner over 10 min. *p<0.05, RM One-way ANOVA, post-hoc Tukey, n = 6 animals. 3 Males/3 females. Data are mean ± SEM.

DOI: https://doi.org/10.7554/eLife.44548.015

provide an additional target by which exogenous nicotine could suppress appetite through extra-hypothalamic neurons.

Chronic hyperactivation of vGlut2$^{BF}$ neurons led to drastic hypophagia, and acute optogenetic activation of these neurons both reduced food intake and caused avoidance of food and food-associated odors. Such dramatic hypophagia has previously been reported in models of lateral hypothalamic lesions, permanent AgRP neuron ablation in the hypothalamic arcuate nucleus, and permanent loss of GABA signaling from AgRP neurons (*Gropp et al., 2005*; *Luquet et al., 2005*; *Tong et al., 2008*; *Wu et al., 2009*). To our knowledge, the current study is the first instance of a model with precisely targeted manipulations to the basal forebrain that leads to hypophagia and starvation. The dramatic phenotype of precipitous weight loss and complete appetite suppression within days of viral mNaChBac expression, indicates the high sensitivity of vGlut2$^{BF}$ neuronal firing. While chronic activation persistent for tens of minutes to days most likely does not occur in nature, changes in appetite especially for neuropsychiatric conditions are often chronic in nature and do not manifest by simply avoiding a single food item once or skipping one meal. Given the sensitivity of this neuronal population and the correlated phenotype, it could be possible that small changes in the excitability of this neuronal population could have chronic phenotypes in altering feeding behaviors. Furthermore, the resistance of the phenotype in leptin deficient mice indicates the feeding pathways altered by our manipulation are either downstream or in parallel to leptin signaling, but capable of overriding leptin deficient induced hyperphagia. Importantly, the hypophagic phenotype seen here is similar to that observed with AgRP neuron ablation, suggesting that the activity of vGlut2$^{BF}$ neurons is a fundamental regulator of feeding behavior (*Gropp et al., 2005*, p. 2005; *Luquet et al., 2005*). Further, moderately obese $Lep^{ob/ob}$ animals survived AgRP neuron ablation (*Wu et al., 2012*), whereas viral mNaChBac manipulation of vGlut2$^{BF}$ neurons in obese $Lep^{ob/ob}$ animals did not rescue prolonged hypophagia and starvation. Accordingly, we and others have identified the lateral hypothalamus as a robust brain target of vGlut2$^{BF}$ neurons involved in feeding regulation (*Do et al., 2016*).

Genetic subpopulations in the lateral hypothalamus (LH) have been extensively studied for their role in feeding related behavior. VGAT-positive inhibitory neurons in the LH have been shown to promote feeding (*Jennings et al., 2015*), while vGlut2 positive neurons in the LH elicit food aversion (*Jennings et al., 2013*). We revealed that vGlut2$^{BF}$ neurons form direct synaptic connections onto vGlut2 positive neurons in the LH, and that their signaling onto LH neurons also diminished food intake and lead to food avoidance, thereby aligning with the previously proposed roles for the lateral hypothalamus. Notably, these findings highlight that a deeper understanding of the connectivity among and between the different cell types within the basal forebrain and lateral hypothalamus promises to reveal more precise mechanisms of how feeding and avoidance behaviors are fully relayed between these two circuit nodes.

An important consideration is the rewarding aspect of consuming food, particularly after a period of fasting. Previous work has shown that lateral hypothalamic projections to the lateral habenula modulate rewarding aspects of food intake, and that basal forebrain projections to the lateral habenula by somatostatin-expressing neurons can also regulate the rewarding aspects of feeding (*Stamatakis et al., 2016*; *Zhu et al., 2017*). The lateral habenula has been implicated in reward-related and aversive behaviors through indirect inhibition of the ventral tegmental area (VTA) dopaminergic neurons, which is another projection site of vGlut2$^{BF}$ neurons. Therefore, given the interconnected nature of cell types between these structures, vGlut2$^{BF}$ neurons could also play a pivotal role in encoding the rewarding aspects of food intake. In future experiments, it will be particularly interesting and important to determine the contribution of vGlut2$^{BF}$ projections to other projection regions, including the lateral habenula and VTA, in controlling the valance of food and/or food-associated odors. Based on cell-body versus LH projection stimulation results, other nodes in this circuit are clearly important since lateral hypothalamic stimulation of vGlut2$^{BF}$ terminals did not fully recapitulate the avoidance behavior seen when vGlut2$^{BF}$ cell bodies were stimulated (*Figures 8*, *9* and *10*). Although unlikely based on this evidence, we can not completely rule out that terminal excitation may backpropagate and activate cell-bodies in the optogenetic experimentation.

Recent investigations in the medial septum (MS) have also revealed neuronal populations that are capable of influencing feeding (*Sweeney et al., 2017*; *Sweeney and Yang, 2016*). Importantly, vGlut2 neurons in the MS reduced food intake through paraventricular hypothalamic projections, but to date, have not been shown to synapse onto lateral hypothalamic neurons (*Sweeney et al.,*

2017). In our study, we found that vGlut2[BF] neurons make functional synapses onto LH neurons and mediate hypophagia in part through this connection. Additionally, anterograde tracing highlighted an absence of vGlut2[BF] neuron innervation to the paraventricular hypothalamus. Excitatory neurons in the MS were also recently identified to influence auditory and somatosensory cue-based avoidance through lateral habenula projections (*Zhang et al., 2018*). That study also identified a unique locomotor phenotype, attributed to MS excitatory neuron projections to the preoptic region. Importantly, when we photostimulated excitatory-neuron cell bodies in the DBB, we did not observe any changes in locomotion, nor did we see increased cellular activity by calcium imaging when head-fixed animals moved on a walking ball. Additionally, our data indicate vGlut2[BF] neurons cause an avoidance phenotype due to food olfactory sensory cues. Together, these studies suggest that vGlut2 neurons in the medial septum and the diagonal band of Broca represent unique basal forebrain populations with distinct projections that mediate appetite suppressing roles.

The robust hypophagia and starvation phenotype seen with the overactivation model described here draws parallels to hypophagic eating disorders. However, it is important to consider the structures investigated in this paper represent subcortical regions, and that neuropsychiatric eating disorders such as anorexia nervosa, binge eating disorders, and bulimia nervosa, are complex multi-faceted conditions involving higher-order cortical regions (*Fuglset et al., 2016*). Additionally, the DBB of the basal forebrain is not considered a primary sensory cortex for olfactory information, and therefore the function of other presynaptic inputs onto vGlut2[BF] neurons need to be further investigated to understand how more complex cognitive states can utilize sensory cues to suppress appetite. Pre-synaptic tracing onto vGlut2[BF] neurons highlight the orbitofrontal and agranular insular cortex as potential targets which could integrate sensory perception and higher level cognition to suppress appetite through basal forebrain circuitry (*Do et al., 2016*).

Sensory perception of food can stimulate or decrease appetite by providing information on a food's palatability, caloric value, and taste. Our results reveal a group of neurons capable of responding to the sensory perception of food, diminishing appetite, and eliciting avoidance to food and food-associated odors. Notably, these findings reveal a previously unknown circuit that links cholinergic signaling in the basal forebrain to feeding behavior, appetite suppression, and food avoidance. In the future, it will be important to uncover potential functions of this circuit in behaviors associated with nicotine-mediated appetite suppression, addiction, and/or learned food aversion associated with eating disorders.

# Materials and methods

**Key resources table**

| Reagent type (species) or resource | Designation | Source or reference | Identifiers | Additional information |
|---|---|---|---|---|
| Strain, strain background (M.Muscularus) | Vglut2-Cre (*Slc17a6-Cre*) | The Jackson Laboratory | RRID:IMSR_JAX:028863 | |
| Strain, strain background (M.Muscularus) | Chat-Chr2 (ChAT-ChR2-EYFP Line 6) | The Jackson Laboratory | RRID:IMSR_JAX:014546 | |
| Strain, strain background (Adeno-Associated Virus Serotype 2/9) | AAV-Ef1a-Flex-hChR2(H134)-EYFP-WPRE-hGHpA (2/9); Flex-Chr2 | This Paper; Neurocconnectivity Core at the Jan and Dan Duncan Neurological Research Institute | | Plasmid subcloned from RRID:Addgene_26973 |
| Strain, strain background (Adeno-Associated Virus Serotype 2/9) | AAV-Ef1a-Flex-hM4Di-mCherry (2/9); Flex-hM4Di-mCherry | This Paper; Neurocconnectivity Core at the Jan and Dan Duncan Neurological Research Institute | | Plasmid subcloned from RRD:Addgene_44362 |

*Continued on next page*

*Continued*

| Reagent type (species) or resource | Designation | Source or reference | Identifiers | Additional information |
|---|---|---|---|---|
| Strain, strain background (Adeno-Associated Virus Serotype 2/DJ8) | AAV-Ef1a-Flex-GCaMP6m-WPRE-hGHpA (DJ8); Flex-GCaMP6m | This Paper; Neurocconnectivity Core at the Jan and Dan Duncan Neurological Research Institute | | Plasmid subcloned from RRID: Addgene_100839 |
| Strain, strain background (Adeno-Associated Virus Serotype 2/DJ8) | AAV-Ef1a-Flex-eGFP-p2a-mNaChBac-WPRE-hGHpA (DJ8); Flex-mNaChBac; mNaChBac | This Paper; Neurocconnectivity Core at the Jan and Dan Duncan Neurological Research Institute | | Plasmid subcloned from plasmid gift from Dr. Mingshan Xue (mxue@bcm.edu) |
| Strain, strain background (Adeno-Associated Virus Serotype 2/DJ8) | AAV-Ef1a-Flex-Synaptophysin::eGFP-WPRE-hGHpA (DJ8); Flex-Synaptophysin | This Paper; Neurocconnectivity Core at the Jan and Dan Duncan Neurological Research Institute | | Plasmid subcloned from RRID:Addgene_73816 |
| Strain, strain background (Adeno-Associated Virus Serotype 2/DJ8) | AAV-Ef1a-Flex-mRuby2 (DJ8); Flex-mRuby2 | This Paper; Neurocconnectivity Core at the Jan and Dan Duncan Neurological Research Institute | | Plasmid subcloned from Addgene:40260 |
| Recombinant DNA reagent (plasmid) | pAAV-Ef1a-Flex-hChR2(H134)-EYFP-WPRE-hGHpA | This paper | RRID:Addgene_26973 | |
| Recombinant DNA reagent (plasmid) | pAAV-Ef1a-Flex-hM4Di-mCherry | This paper | RRID:Addgene_44362 | |
| Recombinant DNA reagent (plasmid) | pAAV-Ef1a-Flex-GCaMP6m-WPRE-hGHpA | This paper | RRID:Addgene_100839 | |
| Recombinant DNA reagent (plasmid) | pAAV-Ef1a-Flex-eGFP-p2a-mNaChBac-WPRE-hGHpA | Plasmid subcloned from plasmid gift from Dr. Mingshan Xue (mxue@bcm.edu) | | |
| Recombinant DNA reagent (plasmid) | pAAV-Ef1a-Flex-Synaptophysin::eGFP-WPRE-hGHpA | This paper | RRID:Addgene_73816 | |
| Recombinant DNA reagent (plasmid) | pAAV-Ef1a-Flex-mRuby2 | This Paper | RRID:Addgene_40260 | |
| Antibody | anti-ChAT (goat monoclonal) | Millipore | Millipore Cat# AB144P; RRID:AB_2079751 | (1: 500 dilution) |
| Antibody | anti-cFos (rabbit polyclonol) | Abcam | Abcam Cat# ab190289; RRID:AB_2737414 | (1:1000 dilution) |
| Chemical compound, drug | CNO (clozapine-n-oxide) | Tocris 4963 | | |
| Commercial assay or kit | Mouse/Rat T4 Elisa | Calbiotech, Inc | T4044T-100 | |
| Commercial assay or kit | Active Ghrelin ELISA | Millipore | EZRGRA-90K | |
| Software, algorithm | Doric Neuroscience Studio | Doric Lenses | | |
| Software, algorithm | MATLAB | Mathworks | RRID:SCR_001622 | |
| Software, algorithm | Optimouse | PMID:28506280 | | |

*Continued on next page*

*Continued*

| Reagent type (species) or resource | Designation | Source or reference | Identifiers | Additional information |
|---|---|---|---|---|
| Software, algorithm | Prism 8 | Graphpad | GraphPad Prism, RRID:SCR_002798 | |
| Software, algorithm | Rodent Liquid diet | Test Diet | Test Diet:101 5LD1 | |
| Other (Mouse Liquid Diet) | Synapse | Tucker-Davis Technologies | Tucker-Davis Technologies; RRID:SCR_006495 | |
| Other (GRIN Lens) | Grin lens | Doric Lenses | Doric: SICL_E_500_80 | |

## Animals

All mice in this study were treated in compliance with US Department of Health and Human Services and Baylor College of Medicine IACUC guidelines. Both male and female mice were used in analyses, littermates were randomly assigned to experimental groups, and further specified below. Mice were between 3–5 months of age for surgeries, and between 3–6 months for experiments. Standard mouse chow (Harlan, 2920X) was used for all experiments, and all animals were maintained on a 12 hr light-dark cycle. Depending on the experiment, mice were either group or singly housed (see below). vGlut2-Cre (*Slc17a6$^{tm2(cre)Lowl}$/J* Stock No. 016963) and Chat-Chr2 (*Chat-COP4\*H134R/ EYFP, Slc18a3 6Gfng/J* Stock No. 014546) mice were originally purchased and are available from Jackson Laboratories. Genotyping for Cre was done using the following primers: forward 5'-GCA TTACCGGTCGATGCAACGAGTGATGAG-3' and reverse 5'-GAGTGAACGAACCTGGTCGAAA TCAGTGCG-3'. Genotyping for Chat-Chr2 was done using the following primers: forward 5'-TCTG TTCCCAGGTCGGCAGC-3' and reverse 5'-GCAAGGTAGAGCATAGAGGG-3'.

## Stereotaxic injections and viral constructs

For all stereotaxic injections and implantation surgeries, mice were anesthetized and maintained under anesthesia using vaporized isoflurane with $O_2$. All injections and implantations were performed using a stereotaxic apparatus synced to Angle Two software for coordinate guidelines. The basal forebrain was targeted through bilateral injections into the horizontal limb of the diagonal band of Broca (from bregma AP= +0.14 mm, DV = −5.8 mm, and ML= ±1.29 mm) or unilateral for calcium imaging experiments. The lateral hypothalamus was targeted using bilateral injections from bregma AP = −1.22 mm, DV = −5.12 mm, ML= ±0.97 mm. Viruses used in experiments include: AAV-Ef1a-Flex-GCaMP6m-WPRE-hGHpA Serotype DJ8, AAV-Ef1a-Flex-eGFP-p2a-mNaChBac-WPRE-hGHpA Sertype DJ8 (sub-cloned from plasmid gift from Dr. Mingshan Xue (mxue@bcm. edu), AAV-Ef1a-Flex-mRuby2 Serotype DJ8, AAV-Ef1a-Flex-Synaptophysin::eGFP-WPRE-hGHpA Serotype DJ8, AAV-Ef1a-Flex-hChR2(H134R)-EYFP-WPRE-hGHpA, serotype 2/9, AAV-Ef1a-Flex-hM4Di-mCherry Serotype 2/9. We injected 200–400 nL of virus for all experiments, except for synaptophysin tracing experiments where 50–100 nL of virus was injected.

## Microscopy and immunohistochemistry

Animals were deeply anesthetized using isoflurane and were transcardially perfused with PBS followed by 4% PFA (Diluted from 16% Paraformaldehyde EM Grade No. 15710 Electron Microscopy Sciences). Brains were dissected and post-fixed in 4% PFA overnight at four degrees Celsius. Brains were cryoprotected in 20% sucrose/PBS solution for 1 day, followed by 30% sucrose/PBS solution for one more day, both at four degrees Celsius. Brains were then embedded and frozen in O.C.T. (Fisher HealthCare No. 4585) and stored at −80 degrees Celsius until being cut. Brains were cut using a cryostat (Leica CM1860) in coronal sections 40 µm. For ChAT and Fos immunohistochemistry, 40 µm free-floating sections were blocked at 1 hr at room temperature in 10% horse serum blocking solution made in PBS-T (1X PBS, 1% Triton-X 100, pH 7.35). Sections were incubated overnight at four degrees Celsius at 1:500 dilution of block solution containing goat anti-ChAT primary antibody (Millipore AB144P) and 1:1000 dilution of rabbit anti-Fos antibody (Abcam ab190289). Sections were washed 4 times for 10 min in plain PBS-T. Sections were then incubated in secondary antibodies

(1:500 donkey anti-goat Alexafluour-488, and donkey anti rabbit Alexaflour-647) each for 2 hr at room temperature. Sections were then washed 5 times at 10 min each in PBS-T. All sections were mounted using DAPI Fluoromount-G (southern Biotech, 0100–20). Sections for synaptophysin tracing were washed with PBS 4 times for 10 min and then mounted with DAPI Fluoromount-G. Detection of fluorescent expression was performed using a Leica TCS SPE confocal microscope, or Leica TCS SP8 STED microscope. For synaptophysin projection quantification all images were taken at 20X holding all imaging parameters constant for each animal. ROIs were drawn in imageJ over target fields and relative intensity was measured. These were normalized on a per animal basis to an ROI over the DBB.

## Neuronal activation after feeding

Male vGlut2-Cre animals were injected with AAV-Flex-mRuby2. Three weeks after viral injection, animals were fasted for 24 hr. They were then placed in individually housed cages for 15 min. One group of animals were presented with standard pellet chow for 1 hr (fed), while a second group of mice lacked chow (fasted group). Mice were then immediately euthanized and processed using the microscopy and immunohistochemistry procedure above. Using identified positions from bregma, BF was identified by borders of cholinergic neuronal populations. Cells within this area were evaluated and counted in order to quantify positively labeled cells.

## In vivo microendoscopy

vGlut2-Cre animals were injected with adeno-associated virus expressing a Cre-dependent fluorescent calcium reporter, GCaMP6M (AAV-Ef1a-Flex-GCaMP6m). For freely behaving experiments, following viral injection, two set screws were drilled into exposed skull tissue. Next, a DORIC GRIN lens (SICL_E_500_80) attached to a protrusion adjustment ring was lowered over the injection site, positioned approximately 80 µm above the virally targeted location. GRIN lenses used had NAs of 0.5, focal distances of 80 µm, 500 µm diameters, and 8.39 mm lengths- adjustable between 5.3 mm to 8.39 mm dependent on the adjustment ring. Dental cement was used to fix the adjustment ring containing the GRIN lens onto the skull. Mice were then allowed to recover for three weeks. Prior to imaging, mice were fasted for 24 hr while having a dummy camera attached to their head. A DORIC microendoscope was next attached to the GRIN lens/protrusion adjustment ring. Mice were then imaged at an exposure rate of 50–250 ms per image, dependent on image quality (4–20 Hz). All results were down-sampled to 4 Hz for freely behaving experiments, and 10 Hz for head fixed animals, since there was better image quality for these experiments. LED power used for imaging was 1–3 mW. During imaging, mice were also video recorded using a Doric high-speed behavior camera at 60fps, which was time-locked with GCAMP imaging using Doric Neuroscience Studio software. For odorant delivery, mice were presented either monomolecular odorants or crushed food pellets dissolved in mineral oil on a cotton swab placed in the corner of a cage. Food interaction was defined as mice approaching and smelling food pellets, but not yet having eaten the food. Physical stimulation of mice was done by gently poking the mice with a cotton swab.

For head fixed experiments, 2 weeks after viral injection a metal head-plate was secured to the mouse skull using dental cement. 24–72 hr after head plate surgeries, mice were attached to a custom head-fix apparatus that includes a running-ball on which mice are able to walk on while their head is affixed to a side bar. The skull hole used for viral injection was re-drilled, and a DORIC GRIN lens (same as above) attached to the microscope was slowly lowered until the basal forebrain was in focus. Odorants were then delivered using a pressurized custom odor delivery robot while basal forebrain responses were imaged. Odorants were delivered for 10 s every 30 s, while being time locked with the beginning of microscope recording. Odors used include: 2-Methylbutylamine (241407 Sigma) dissolved in 5 mL mineral oil to a concentration of 4.5 µM, 100 µL Butyric Acid (103500 Sigma) dissolved in 5 mL mineral oil, food pellets were sonicated in mineral oil, and mineral oil alone. Gently touching mice with a cotton swab resulted in mice running on the wheel and was used to measure calcium activity from physical movement. Responses were averaged from two trials of randomized odor delivery.

## Calcium image processing

Images, if necessary, were corrected for motion artifacts using DORIC neuroscience studio image analyzer. For freely moving animals, cells were identified by direct visualization due to accurate focusing. For head-fixed recordings, a combination of PCA cell identification and hand drawn ROIs were utilized to isolate cells in the DORIC neuroscience studio software. The software calculated Delta F/F for whole-field images and ROIs. Traces were extracted, and then further analyzed using existing plotting functions in MATLAB, including plot and images. For freely behaving animals, DF/F traces were normalized for individual animals from 0 to 1 using the animals' own maximum and minimum DF/F trace values. These normalized traces were then averaged across animals in order to compare activation. Cells were considered active if they achieved at least a two standard deviation increase in response for freely behaving animals, and three standard deviation increase in response for head-fixed animals.

## Electrophysiology

For all slice experiments, animals were anesthetized with isoflurane and perfused with cold artificial cerebrospinal fluid (ACSF) solution pH 7.35 mOsm 305–315 containing: 125 mM NaCl, 2.5 mM KCl, 1.25 mM $NaH_2PO_4$-$H_2O$, 2 mM $CaCl_2$, 1 mM $MgCl_2$-$6H_2O$, 20 mM Glucose, 25 mM $NaHCO_3$. Brains were rapidly removed and transferred into sucrose-based cutting solution pH 7.35 containing: 87 mM NaCl, 2.5 mM KCl, 1.25 mM $NaH_2PO_4$-$H_2O$, 0.5 mM $CaCl_2$, 7 mM $MgCl_2$-$6H_2O$, 13 mM Ascorbic Acid, 75 mM Sucrose, 10 mM Glucose, 25 mM $NaHCO_3$ and continuously bubbled with 5% $CO_2$/ 95% $O_2$. 300 µm thick coronal brain slices were prepared using a Leica VT1200 vibratome and placed in recover for 15 min at 37 degrees Celsius in 5% $CO_2$/95% $O_2$ bubbled ACSF solution. They were then gradually lowered to room temperature 25 degrees Celsius for and allowed to acclimate for at least 15 min before recording. For recording, slices were transferred into a recording chamber continuously perfused at 1–2 mL/min at 25 degrees Celsius. Neurons were identified by transmitted light DIC and fluorescent imaging (BX50WI, Olympus). Recordings were obtained using an Axon MultiClamp 700B amplifier digitized at 10 kHz (Axon Digidata 1440A). Recording electrodes (3–5 megaohms) were fabricated from borosilicate glass microcapillaries (outer diameter, 1.5 mm) with a micropipette puller (Sutter Instruments). For current clamp recordings, a potassium gluconate internal solution was used consisting of 120 mM K-gluconate, 5 mM KCl, 2 mM $MgCl_2$-$6H_2O$, 2 mM Mg-ATP, 10 mM $Na_2$ phostphocreatine, 0.4 mM Mg-GTP, 10 mM HEPES EGTA, 0.05 mM EGTA, pH 7.3 with KOH, 290 mOsm. For voltage clamp recordings, internal solution containing the following was used: 120 mM Cs-methanesulfonate, 2 mM $MgCl_2$-$6H_2O$, 0.05 mM $CaCl_2$, 6 mM CsCl, 20 mM HEPES, 0.2 mM EGTA, 10 mM phosphocreatine di(Na) salt, 4 mM ATP-Mg, 0.4 mM GTP-Na, pH 7.2 with CsOH, and 290-300mOsm with CsMeSO₄. *mNaChBac Recording:* Brain slices containing the basal forebrain were prepared from 12 to 16 week-old animals expressing eGFP-p2a-mNaChBac in one hemisphere, and mRuby2 in the contralateral hemisphere. Resting potential was recorded within seconds of breaking into a cell. Cells with a series resistance of less than 30 megaOhms and <25% change for duration of the experiment were used for analysis. Analysis of current-clamp electrophysiological data was performed using pClamp10 (Molecular Devices). Single action potential parameters were measured at threshold, and action potential amplitudes were measured relative to action potential threshold.

## Channelrhodopsin electrophysiology

Brain slices containing either the basal forebrain or lateral hypothalamus were prepared from 12 to 16 week-old animals. Cells were patched using voltage clamp internal solution. Cells were recorded for a minimum of 10 sweeps for each pharmacological condition. Each sweep consisted of 2 s of recording, with an excitation duration of 10 ms of blue light (470 nm) exposure with 30 s between each sweep. For evaluation of cholinergic connectivity onto VGlut2 +basal forebrain neurons, we used ACSF that had the following final pharmacological concentrations in the slice chamber: 20 µM AP-5 (Tocris), 10 µM CNQX (Tocris), 2 µM GABAzine (Tocris), 10 µM Atropine (Sigma), 1 µM TTX (Tocris), 0.5 µM 4-AP (Tocris), and/or 0.5 mM Mecamylamine (Tocris). For lateral hypothalamus recordings, ACSF with the following pharmacological concentrations in the slice chamber were used: 1 µM TTX (Tocris, 0.5 µM 4-AP (Tocris), 20 µM AP-5 (Tocris), 10 µM CNQX (Tocris).

## In vivo extracellular recordings

12–16-week-old vGlut2-Cre +/+mice were injected with either Flex-Chr2 or with both Flex-Chr2 and Flex-NaChBac. During the same surgery a metal head-plate was secured to the mouse skull using dental cement. 24–72 hr after head plate surgeries, mice were attached to a custom head-fix apparatus that includes a running-ball on which mice are able to walk on while their head is affixed to a side bar. The skull hole used for viral injection was re-drilled. A 32-channel NeuroNexus Optrode was stereotaxically lowered into the brain to the depth of the basal forebrain. Cells were identified by applying 10 Hz light stimulation. Data for spontaneous frequency was recorded for 5 min, ending with a 10 Hz photostimulation using a 465 nm laser (DORIC). Data were recorded using a multichannel Tucker-Davis Technologies (TDT) system (Alachua, Florida), and single unit spikes were isolated using a 3.0 standard deviation cut-off from noise and a Bayesian clustering algorithm pre-built into the TDT Synapse software. Data were exported into MATLAB and further quantified to obtain spontaneous frequencies.

## vGlut2 basal forebrain neuron activation with mNaChBac Assay

12–16 week-old equal number of male and female vGlut2-Cre +/- mice were separated into individually housed cages for 1 week, and mice and chow were measured daily to establish average baseline weight and daily food intake. After 1 week, mice were injected with either Flex-eGFP-p2a-mNaChBac or Flex-GFP as described above. Mice and food were continuously measured daily. If during the daily mouse weighing, the mice lost more than 20% of baseline weight, the final weight was recorded, and the mouse was humanely sacrificed per IACUC guidelines. Additionally, if a mouse displayed anorexia for greater than 3 days, it was also sacrificed.

## Liquid diet and oral gavage assay

12–16 week-old VGlut2-Cre +/- mice were separated into individually housed cages and transitioned to a liquid diet 1 week prior to viral injection by placing both chow and liquid diet (Test Diet Rodent liquid diet 101 5LD1) in mouse water bottles. After viral injection, mouse food liquid diet intake, and body weight were measured twice daily. Based on manufacturer guidelines that mice should consume at least 20 g of liquid diet, any difference between actual intake and this minimum, more was administered as oral gavage twice daily to maintain mouse weight higher than IACUC sacrifice policy of 25% of baseline weight loss.

## Hormone analysis

12–16 week-old VGlut2-Cre +/- mice were injected with either Flex-eGFP-p2a-mNaChBac-or Flex GFP virus. Blood was obtained using lancet puncture of the submandibular vein 4 days post viral injection. Blood was collected into 200 μL lavender top EDTA tubes (microvette 200 K3E). Tubes were centrifuged at 2500 g at room temperature for 5 min. Supernatant plasma was immediately isolated, flash frozen on dry ice, and stored at −80 degrees. Mouse pituitary panels were measured by the Vanderbilt Hormone and Analytics Core using a Mouse Pituitary Luminex Panel. Mouse free T3 was measured by the Vanderbilt Hormone and Analytics Core using a free T3 radioimmunoassay kit. Mouse T4 was measured using a Mouse/rat T4 ELISA (Calbiotech, Inc). Mouse insulin, glucose, and leptin were measured by the Baylor College of Medicine Mouse Metabolic and Phenotyping Core. Ghrelin levels were measured using a Rat/Mouse Active Ghrelin ELISA kit (Millipore EZRGRA-90K).

## Chemogenetic hM4Di assay

12–16 week-old, three male, two female VGlut2-Cre mice were injected bilaterally into basal forebrains with an hM4D expressing AAV. After two weeks of recovery, mice were separated and individually housed for 1 week and subjected to overnight fasting or ad-libitum feeding in their home cage. For control experiments mice were injected i.p. with sterile saline, for experimental conditions mice were injected i.p. with CNO (5 mg/kg mouse body weight). 30 min after injection, food was presented to animals and measured over the course of 5 hr. Trials were randomized, separated 1 week apart, and conducted on the same animals. Paired statistics were utilized to compare experimental and control conditions.

## Optogenetic food intake assay

12–16-week-old mice were injected bilaterally into the basal forebrain with Flex-Chr2 or Flex GFP virus as described above. Two weeks after viral injection, mice were bilaterally implanted with 200 µm silica fiber optic implants made in-house (Thor Labs) with 230 µm attachment ferrules (Precision Fiber Products), and situated. 1 mm above the viral injection sites in the basal forebrain and projection site in the lateral hypothalamus. Fiber optic implants were held in place by a cap made from adhesive cement (C and B Metabond Dental cement Parkell). On top of which a crosslinked flash acrylic (Yates-Motloid 44115 and 44119) cap was constructed. Mice were allowed another 2 weeks for recovery. Mice were acclimated to a behavior chamber for 30 min on day 1. On day two mice, were acclimated again while tethered to a dual fiber optic cord (Doric Lenses). Mice were subsequently fasted over-night. On day 3, mice were placed in behavior chambers while tethered to the dual fiber optic cord. One group was photostimulated at 5 Hz (10 ms pulses,~4 mW) for 30 min over the basal forebrain, another group photostimulated over the lateral hypothalamus, and one group had no photostimulation. Food was placed on a petri dish in one corner of the cage. After 30 min food was measured. This three-day experiment was repeated twice more with 1 week between the start of each experiment, as the mice randomly switched photostimulation groups (Basal forebrain, lateral hypothalamus, or none).

## Variable frequency optogenetic stimulation assay

12–16 week-old mice were bilaterally injected into the basal forebrain with Flex-Chr2 as described above. Two weeks after viral injection, mice were bilaterally implanted with 200 µm silica fiber optic implants made in-house (Thor Labs) with 230 µm attachment ferrules (Precision Fiber Products) and situated. 1 mm above the viral injection sites in the basal forebrain. Fiber optic implants were held in place by a cap made from adhesive cement (C and B Metabond Dental cement Parkell). On top of which a crosslinked flash acrylic (Yates-Motloid 44115 and 44119) cap was constructed. Mice were allowed another 2 weeks for recovery. Mice were acclimated to a behavior chamber for 30 min on day 1. On day 2, mice were acclimated again while tethered to a dual fiber optic cord (Doric Lenses). Mice were subsequently fasted over-night. On day 3, mice were placed in behavior chambers while tethered to the dual fiber optic cord. Mice were photostimulated at variable frequencies 0–20 Hz (10 ms pulses,~4 mW) for 10 min over the basal forebrain. This was repeated weekly with mice fasted overnight prior to the experiment. Photostimulation frequencies included 0, 5, 10, and 20 Hz. Preweighed food was placed on a petri dish in one corner of the cage. After 10 min, food weight was measured.

## Optogenetic food pellet avoidance assay

Food pellets were placed in three random corners of a behavior chamber, leaving one corner devoid of food. Mice were recorded using a DORIC high speed behavior camera. The same mice from the food intake assay were used for these experiments in the same behavior chambers, so no re-acclimation was performed. On day 1, mice were separated into either BF stimulation, LH stimulation, or no photo-stimulation groups. They were fasted overnight prior to experiments. Mice were photostimulated at 5 Hz (10 ms pulses,~4 mW) for the 30 min experimental timeframe. Experiments were repeated twice more, one week apart from each other as mice switched groups between stimulation locations. Unstimulated groups were still tethered to the fiber optic cord. Videos of mice were analyzed using open source software OptiMouse in MATLAB (*Ben-Shaul, 2017*).

## Optogenetic food smell avoidance assay

The same mice from food intake and food pellet avoidance assays were used for these experiments in the same behavior chambers. Chow was spiked in mouse home cages with 1 mL of R-limonene dissolved in mineral oil for 2 weeks to form a passive association of R-limonene with food pellets. After 2 weeks, mice were fasted overnight. On the day of experiments, mice were placed in behavioral chambers and attached to fiber optic cords, either to the basal forebrain or lateral hypothalamus. For half of the animals, a q-tip with R-limonene dissolved in mineral oil was placed in an open 1.5 mL tube in three corners, leaving one corner empty. Mice spent 10 min in this situation. Next, 5 Hz photostimulation (10 ms pulses,~4 mW) was presented for 10 min. Next, the q-tips/Eppendorf tubes with R-limonene were swapped with q-tips/Eppendorf tubes containing S-limonene dissolved

in mineral oil as controls. Mice were recorded in this situation for 10 min while being photostimulated. The other half of the animals went through the sequence in reverse (s-limonene with photostimulation, r-limonene with photostimulation, r-limonene with no photostimulation). This experimental paradigm was repeated one week later, as mice switched from having their basal forebrain to lateral hypothalamus photostimulated, or vice versa.

## Quantification and statistics

Data were analyzed using GraphPad Prism seven for Mac OSX. For calcium imaging studies, a minimum of five animals with at least two animals of each sex were used. Active neurons were identified with fluorescent activity two standard deviations above baseline activity for freely behaving animals, and three standard deviations for head-fixed animals. Graphs with single time points were analyzed using repeated measures one way Anova tests, with Tukey's post hoc multiple comparison sets. Graphs with multiple time points, 2way ANOVA repeated measures were performed, with multiple comparisons using Tukey post hoc correction. For comparison of means between two groups, two tailed student's t-tests were used. No statistical methods were used to predetermine sample size. When relevant, randomization was carried out. Behavioral experiments were randomized. There was blinding during initial allocation of animals into groups, but not thereafter. All cell counts were blinded.

## Contact for reagent and resource sharing

Further information and requests for resources and reagents should be directed to and will be fulfilled by the Lead Contact, Benjamin R Arenkiel (arenkiel@bcm.edu).

## Acknowledgements

We would like to thank Dr. Huda Zoghbi, and Dr. Yong Xu for their helpful input and critical comments towards preparing this manuscript. The project described was supported through the multi-PI award R01DK109934 to BRA and QT, and the Neuroconnectivity Core at Baylor College of Medicine, which is supported by IDDRC Grant Number 1 U54 HD083092 from the Eunice Kennedy Shriver National Institute of Child Health and Human Development. The content is solely the responsibility of the authors and does not necessarily represent the official views of the Eunice Kennedy Shriver National Institute of Child Health and Human Development or the National Institutes of Health. The project was supported by F30DK112571 to JMP. The project described was in part also supported by the Vanderbilt Hormone and Analytics Core supported by NIH grants DK059637 (MMPC) and DK020593 (DRTC). We would also like to thank the McNair Medical Institute and Charif Souki for their ongoing and generous support.

## Additional information

### Funding

| Funder | Grant reference number | Author |
|---|---|---|
| National Institute of Diabetes and Digestive and Kidney Diseases | F30DK112571 | Jay M Patel |
| National Institute of Neurological Disorders and Stroke | R01NS078294 | Jennifer Selever Benjamin R Arenkiel |
| Eunice Kennedy Shriver National Institute of Child Health and Human Development | U54HD083092 | Jennifer Selever Benjamin R Arenkiel |
| National Institute of Diabetes and Digestive and Kidney Diseases | R01DK109934 | Benjamin R Arenkiel Qingchun Tong |

The funders had no role in study design, data collection and interpretation, or the decision to submit the work for publication.

## Author contributions
Jay M Patel, Conceptualization, Software, Formal analysis, Funding acquisition, Validation, Visualization, Methodology, Writing—original draft, Writing—review and editing; Jessica Swanson, Validation, Investigation, Writing—review and editing; Kevin Ung, Investigation, Writing—original draft, Writing—review and editing; Alexander Herman, Conceptualization, Methodology, Writing—original draft, Writing—review and editing; Elizabeth Hanson, Investigation, Writing—review and editing; Joshua Ortiz-Guzman, Investigation, Methodology; Jennifer Selever, Resources, Funding acquisition, Project administration; Qingchun Tong, Conceptualization, Resources, Supervision, Funding acquisition, Investigation, Methodology, Writing—original draft, Project administration, Writing—review and editing; Benjamin R Arenkiel, Conceptualization, Resources, Supervision, Funding acquisition, Investigation, Visualization, Methodology, Writing—original draft, Project administration, Writing—review and editing

## Author ORCIDs
Jay M Patel (iD) http://orcid.org/0000-0003-2746-5716
Benjamin R Arenkiel (iD) http://orcid.org/0000-0001-9047-2420

## Ethics
Animal experimentation: All mice in this study were treated in compliance with US Department of Health and Human Services and Baylor College of Medicine and in strict accordance with the recommendations in the Guide for the Care and Use of Laboratory Animals of the National Institutes of Health. All of the animals were handled according to approved institutional animal care and use committee (IACUC) protocol (AN5966) at Baylor College of Medicine. All surgery was performed in accordance with the approved protocol, and every effort was made to minimize suffering.

## Decision letter and Author response
Decision letter https://doi.org/10.7554/eLife.44548.018
Author response https://doi.org/10.7554/eLife.44548.019

## Additional files

### Supplementary files
• Transparent reporting form
DOI: https://doi.org/10.7554/eLife.44548.016

### Data availability
All data generated or analysed during this study are included in the manuscript and supporting files.

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
