## [Decision Letter]

Thank you for submitting your article "Sensory perception drives food avoidance through excitatory basal forebrain circuits" for consideration by *eLife*. Your article has been reviewed by three peer reviewers, including Richard D Palmiter as the Reviewing Editor and Reviewer #1, and the evaluation has been overseen by Catherine Dulac as the Senior Editor. The following individuals involved in review of your submission have agreed to reveal their identity: Zachary A Knight (Reviewer #2).

The reviewers have discussed the reviews with one another and the Reviewing Editor has drafted this decision to help you prepare a revised submission.

This paper describes a series of experiments showing that glutamatergic neurons in the basal forebrain project onto glutamatergic neurons in the lateral hypothalamus and activation of these neurons can robustly inhibit food intake and can lead to starvation. On the other hand, inactivation of these neurons promotes food intake and weight gain. The authors also show that these neurons are activated by food odors and odor cues that have been associated with food.

While there is general enthusiasm for this study, there are a few major points that need to be addressed. There are more details in the individual reviews (below).

1) The relationship between activation of these neurons by food odors and anorexia needs to be explained because the smell of food does not normally lead to a loss of appetite, but rather has the opposite effect. Linking activation of these basal forebrain neurons to conditions that normally promote anorexia is needed.

2) If the first point can be addressed, then better analysis of imaging data is needed as described by the reviewers.

3) A more thorough explanation of how the bacterial sodium channel leads to hyperactivity of the neurons is needed. Figure 2C shows that the firing frequency is only 0.04 Hz but action potential width is increased 100 fold. Is that sufficient for the robust behavioral phenotype? Inclusion of data comparing spike frequency/current injection for these neurons would help. Inclusion of an alternative activation paradigm (e.g. expression of hM3Dq and chronic CNO delivery) would also help solidify the chronic-activation results.

4) A more thorough analysis of the axonal projections of the basal forebrain neurons is required.

*Reviewer #1:*

The authors of this study show that chronic activation of VGluT2+ neurons in basal forebrain (Diagonal Band of Broca, DBB) results in severe anorexia and starvation. Transient activation of these neurons (ChR2) or their projections onto VGluT2+ neurons in the lateral hypothalamus inhibits food intake. Conversely, inhibiting these neurons promotes food intake. These DBB are activated by food odors, but not by eating per se, and they can be activated by novel odors that have previously been paired with food. When the DBB neurons are activated, mice avoid the region of a box where food is present, suggesting that food odor is aversive when these neurons are activated.

1) The authors start their Introduction with the concept that animals are attracted to safe food odors but repelled by odors associated with spoiled food. Thus, after knowing that these neurons inhibit feeding, I was expecting them to show that DBB neurons are selectively activated by odors associated with spoiled food. However, it appears that they are activated by natural food odors and unnatural odors that have been associated with normal food. There is no doubt that these neurons are activated by food-related odors, but they are probably activated by all sorts of other stimuli, some of which one might expect to produce food avoidance, e.g. a wide variety of threatening or nausea-inducing conditions. Determining whether the DBB VGluT2+ neurons are activated by conditions that are known to promote anorexia should be included in this study.

2) In Figure 2, the authors show that chronic activation of DBB neurons (with the bacterial sodium channel) results in severe anorexia. The authors show that this strategy activates neurons, but it is difficult to decipher how that translates into firing rate in vivo. If the authors could determine the firing rate, that would make it easier to interpret the experiments in Figure 5, where the authors show that 5-Hz ChR2 stimulation also inhibits food intake (~ 50%). A dose-response experiment with increasing frequency of ChR2 stimulation would help the reader know whether robust activation can lead to complete cessation of feeding (like the chronic experiments).

3) The authors show projections of the DBB Vglut2 neurons to the LH, but I suspect that they project their axons to many other brain regions as well. A supplemental figure showing all the projections should be included. Showing that activation of one (or more) of the other projections has no effect on food intake would be a nice addition.

*Reviewer #2:*

The manuscript investigates the role of VGluT2+ neurons in basal forebrain (BF) in regulation of food intake. The authors show using miniscope calcium imaging that a subset of Vglut2-BF neurons are activated when mice interact with a food pellet or smell a food associated odor. Chronic activation of these cells via overexpression of a bacterial sodium channel results in hypophagia and starvation in wild-type and ob/ob mice, whereas silencing using hM4D increases food intake. Optogenetic stimulation of Vglut2-BF neurons or their projections to LH decreases food intake and causes mice to avoid places containing food or food associated odors.

This is an interesting paper that builds on recent work implicating GABAergic and cholinergic cell types in BF in the regulation of food intake by showing that VGLUT2-BF neurons can also have potent effects on feeding. A strength of the paper is the use of several different approaches to bidirectionally manipulate neural activity, resulting in consistent effects on food intake and body weight with big effect sizes. The major weakness is the analysis of the calcium imaging data and the logical connection between the dynamics and the results of manipulation experiments. These points can be addressed mostly with new analysis and discussion.

1) The miniscope data should be presented and analyzed in way that makes it clearer what exactly the neurons are encoding. To start, the authors should show in Figure 1 the total number of neurons that were imaged in BF and the percentage of neurons that were odor responsive. They should also report the percentage of non-odor responsive cells that showed responses to any of the other stimuli tested. Currently it is unclear how the cells in Figure 1E-G were selected or what the population distribution of activity of VGLUT2 neurons looks like.

Beyond this, Figure 1 gives the impression that Vglut2-BF neurons are activated transiently by the first presentation of a food related odor, and then remain silent throughout the course of the ensuing meal. Is this true? The authors should present some miniscope data recorded during natural behavior that are plotted on the timescale of a meal, rather a 10 second window around events. Do the neurons respond to repeated presentation of the same food odor? How long does the response to "food interaction" last, and how is this so cleanly distinguished from "food eating" (Figure 1E), given that these are overlapping behaviors, and that the mouse continues to smell the food while eating? The authors suggest that these responses do not depend on whether the animal is fasted or fed, but they should show the data that support this. They should also explain what the arrows mean in Figure 1E and how the y-axis is defined in Figure 1F,G.

The authors should clarify the specificity of the food-odor responses. The text states that "we did not observe activation of Vglut2-BF neurons to non-food associated odors, novel odorants" but Figure 1E shows responses to novel odorants (yellow trace). These responses are of similar magnitude to the food odor response but slightly delayed. What does this mean?

2) After clarifying the response properties of VGLUT2-BF neurons, the authors should outline their model for what these dynamics encode. For example, do they signal merely the presence of food in the environment or something broader related to salience, reward, etc. They should also explain in the discussion how this relates the broader literature about the function and in vivo activity of BF neurons.

3) Finally, the authors should explain how the manipulation data and natural dynamics are logically connected. This is important because it seems like the neurons are activated by the smell of food, but the smell of food does not inhibit food intake. Likewise given that these neurons are activated only transiently (a few seconds), it is unclear how to interpret feeding phenotypes that result from chronic manipulations that persist for tens of minutes to days. Does this ever happen naturally? The Discussion section would be better used to highlight these tensions and unresolved questions and make the interpretation clearer.

4) Figure 7 shows that stimulation of an anorexia circuit can lead animals to avoid places with food or food odors, which make sense, but this does not explain why the smell of food would activate an anorexia circuit (Figure 1). The authors should clarify this by rewriting the setup for Figure 7 (subsection “Acute activation of Vglut2BF neurons and their LH projections drives avoidance of food odors”), which at present seems to imply that Figure 7 is a test of the logical implications of Figure 1.

*Reviewer #3:*

In this manuscript by Patel et al., the authors identify and characterize a circuit originating from Vglut2 expressing neurons in the Basal Forebrain that project to the Lateral Hypothalamus to negatively regulate feeding behavior. The main results are that BF Vglut2 cells are activated by food odor or food interaction. Direct long-term activation of Vglut2 BF neurons via introduction of a constitutively active bacterial sodium channel results in hypophagia, weight loss, and starvation. BF Vglut2 cells receive (some) input from BF Cholinergic cells, and project out to the LH where they synapse onto glutamatergic neurons. Photostimulation of these fibers directly in the LH also reduced food intake and causes mice to avoid locations paired with food odors. Overall, the experiments are novel and well executed. The manuscript is also easy to follow and well written.

1) It would be helpful if more zoomed out confocal images showing the extent of viral labeling in the BF were shown. Are there cells outside of the BF that were labeled? Additionally, zoomed out images of fibers in LH would be helpful in order for the reader assess how specific this projection is for targeting the LH. Where other terminal fields (aside from the LH) discovered following labeling of Vglut2 BF neurons?

2) Are the data in Figure 1E trial averaged or example data from each neuron in only one trial?

3) Data in Figure 5C and D appear like they should be combined and analyzed with a two-way ANOVA. Same issue with Figure 6B.

4) For microendoscopic imaging experiments, please report the optical specs for the GRIN lenses used. Including NA, focal distance, and GRIN lens length. Additionally, please report LED power parameters in mW.

5) How was food interaction quantified? This should be clarified.

6) Discussion section, please clarify what particular gross hypothalamic lesions reduce feeding as some hypothalamic lesions (VMH lesions) can increase feeding.

---

## [Author Response]

[…] Reviewer #1:The authors of this study show that chronic activation of VGluT2+ neurons in basal forebrain (Diagonal Band of Broca, DBB) results in severe anorexia and starvation. Transient activation of these neurons (ChR2) or their projections onto VGluT2+ neurons in the lateral hypothalamus inhibits food intake. Conversely, inhibiting these neurons promotes food intake. These DBB are activated by food odors, but not by eating per se, and they can be activated by novel odors that have previously been paired with food. When the DBB neurons are activated, mice avoid the region of a box where food is present, suggesting that food odor is aversive when these neurons are activated.1) The authors start their Introduction with the concept that animals are attracted to safe food odors but repelled by odors associated with spoiled food. Thus, after knowing that these neurons inhibit feeding, I was expecting them to show that DBB neurons are selectively activated by odors associated with spoiled food. However, it appears that they are activated by natural food odors and unnatural odors that have been associated with normal food. There is no doubt that these neurons are activated by food-related odors, but they are probably activated by all sorts of other stimuli, some of which one might expect to produce food avoidance, e.g. a wide variety of threatening or nausea-inducing conditions. Determining whether the DBB VGluT2^+^ neurons are activated by conditions that are known to promote anorexia should be included in this study.

We agree with the reviewer that the original manuscript did not adequately address this key conceptual point. With new experimentation (revised Figure 5), we presented mice with 2 separate classes of known aversive odorants while recording responses. Indeed, we found strong activation of vGlut2 basal forebrain neurons to aversive odorants that (1) activated a broader population of cells, and (2) more strongly activated similar cells as food odors. Together, these data provide a stronger conceptual framework as to why activation of these neurons leads to anorexia. We thank the reviewer for this impactful suggestion to bolster the findings and add conceptual rationale for the observed hypophagic phenotype associated with targeted activation.

*2) In Figure 2, the authors show that chronic activation of DBB neurons (with the bacterial sodium channel) results in severe anorexia. The authors show that this strategy activates neurons, but it is difficult to decipher how that translates into firing rate* in vivo*. If the authors could determine the firing rate, that would make it easier to interpret the experiments in Figure 5, where the authors show that 5-Hz ChR2 stimulation also inhibits food intake (~ 50%). A dose-response experiment with increasing frequency of ChR2 stimulation would help the reader know whether robust activation can lead to complete cessation of feeding (like the chronic experiments).*

We have performed in vivo extracellular recordings in NaChBac injected animals and revealed increased firing rates of Vglut2 neurons in the basal forebrain. These data corroborate the low firing frequency seen in slice culture and show an increase in firing rate with NaChBac in a similar range to the original 5Hz Chr2 stimulation. These data are now highlighted in a new Figure 2—figure supplement 1D. We have also performed a dose-response experiment with increasing frequencies of ChR2 stimulation to compare with the chronic excitation experiments, since the neuronal activation properties of Chr2 vs NaChBac are different. Interestingly, we found increasing frequencies to 20 Hz reduced food intake even further. These data are now included in a new Figure 8D. In whole cell recording experiments in slices, current injections showed increased area under the curve, indicating longer activation of NaChBac expressing neurons per current injected. These new data are now represented in Figure 2—figure supplement 1C.

3) The authors show projections of the DBB Vglut2 neurons to the LH, but I suspect that they project their axons to many other brain regions as well. A supplemental figure showing all the projections should be included. Showing that activation of one (or more) of the other projections has no effect on food intake would be a nice addition.

We have added extensive new data (Figure —figure supplement 1) showing projections from DBB Vglut2 neurons to multiple other target regions. Since these outputs are diverse, and that we are currently investigating subsets of these outputs towards potential roles feeding and other types of behaviors, we have decided to focus here on LH outputs that we could clearly confirm previously unknown roles. Future experimentation beyond this study will uncover if these outputs act synergistically to regulating feeding, or if they each convey unique features.

Reviewer #2:[…] 1) The miniscope data should be presented and analyzed in way that makes it clearer what exactly the neurons are encoding. To start, the authors should show in Figure 1 the total number of neurons that were imaged in BF and the percentage of neurons that were odor responsive. They should also report the percentage of non-odor responsive cells that showed responses to any of the other stimuli tested. Currently it is unclear how the cells in Figure 1E-G were selected or what the population distribution of activity of VGLUT2 neurons looks like.

We now present these additional data in a revised Figure 1 showing the total number of neurons imaged in the basal forebrain, the percentage that were odor responsive, and the percentage that did not respond to odors and/or responded to other stimuli (walking, grooming). We also now provide heat-map data of all the cells recorded from one animal organized by type of activation, showing population distribution.

Beyond this, Figure 1 gives the impression that Vglut2-BF neurons are activated transiently by the first presentation of a food related odor, and then remain silent throughout the course of the ensuing meal. Is this true? The authors should present some miniscope data recorded during natural behavior that are plotted on the timescale of a meal, rather a 10 second window around events.

First of all, to address the reviewer’s inquiry, in all recordings of both freely-behaving and head-fixed animals, these neurons were mostly silent during the recording session. We routinely observe this in the basal forebrain devoid of sensory stimulation, but most cell types in the DBB become much more active with presentation of specific sensory cues or stimuli. To incorporate the reviewer suggestion, we now provide longer traces of cell responses, annotated with behavior to show that these neurons are activated by specific stimuli for both freely behaving and head-fixed animals. These new data are now presented in in Figure 1—figure supplement 1D and Figure 5A, respectively.

Do the neurons respond to repeated presentation of the same food odor? How long does the response to "food interaction" last, and how is this so cleanly distinguished from "food eating" (Figure 1E), given that these are overlapping behaviors, and that the mouse continues to smell the food while eating?

This is a good point, and we now provide new experimental data showing how neurons respond to repeated presentations of the same food odor, and/or to repeated trials to aversive odors. We have provided longer example traces to qualitatively illustrate this in Figure 1—figure supplement 1D and Figure 5A. Moreover, times associated with food interaction was annotated as time spent investigating food until mice began to eat. We also noted that these same neurons reduce activity upon consumption of food. While these data do not allow us to rule out that animals continue to perceptually smell food while eating, we hypothesize there is a lack of attentive investigation or active pursuit when the mouse is eating, thus leading to inhibition of these neurons. Although this remains to be directly tested, we observe a similar response in some Vglut2 neurons when mice go from a moving to resting state as well.

The authors suggest that these responses do not depend on whether the animal is fasted or fed, but they should show the data that support this.

We agree that this point was confusing, and have since removed the claim from the manuscript given the lack of conclusive supporting data.

They should also explain what the arrows mean in Figure 1E and how the y-axis is defined in Figure 1F, G.

For clarification to the reviewer, the y-axis represents normalized percent DF/F. Each animal’s percent DF/F was normalized to its own maximum and minimum to establish a range between 0-1. We noted significant variability in the maximum DF/F for each animal due to slight variations in GRIN Lens placement in freely behaving animals. We found normalizing the signal for each animal allowed for a more accurate comparison since with the DORIC system, the GRIN lenses are not adjustable after implantation. For head-fixed animals, where we were able to lower the lens to an optimal focal plane, all corresponding data are presented as DF/F, or an average DF/F between multiple mice. The arrows in the original figure were to indicate that each row corresponded to the same cell across heat maps. We have now removed the arrows, and included this description in the figure legend to avoid confusion.

The authors should clarify the specificity of the food-odor responses. The text states that "we did not observe activation of Vglut2-BF neurons to non-food associated odors, novel odorants" but Figure 1E shows responses to novel odorants (yellow trace). These responses are of similar magnitude to the food odor response but slightly delayed. What does this mean?

We acknowledge and agree that the description in the original manuscript was confusing and thank the reviewer for attention to this point. Considering new data presented in the revised manuscript, we now show how different odorants differentially activate basal forebrain excitatory neurons. Notably, we found aversive odorants to be strong and differential activators of Vglut2-BF neurons. These data are now presented in a modified Figure 5. In light of these new data, we have removed the statements associated with “non-food associated odors”, since Vglut2-BF neurons indeed appear to be differentially activated by various odors.

*2) After clarifying the response properties of VGLUT2-BF neurons, the authors should outline their model for what these dynamics encode. For example, do they signal merely the presence of food in the environment or something broader related to salience, reward, etc. They should also explain in the discussion how this relates the broader literature about the function and* in vivo *activity of BF neurons.*

We have added discussion as to what the dynamics of Vglut2-BF neurons may potentially encode, how this relates to the broader literature, and previous work investigating BF neuron activity. Briefly, it appears that differences in the number of neurons being activated, and strengths of their activation, dynamically change between types of odorants. This may indicate that valence is being assigned to odorant cues upstream, that are then relayed downstream to behavioral output circuits in the lateral hypothalamus and/or other downstream targets. However, this remains to be directly tested.

3) Finally, the authors should explain how the manipulation data and natural dynamics are logically connected. This is important because it seems like the neurons are activated by the smell of food, but the smell of food does not inhibit food intake. Likewise given that these neurons are activated only transiently (a few seconds), it is unclear how to interpret feeding phenotypes that result from chronic manipulations that persist for tens of minutes to days. Does this ever happen naturally? The Discussion section would be better used to highlight these tensions and unresolved questions and make the interpretation clearer.

Additional experiments and new data presented indeed reveal that Vglut2-BF neurons are robustly activated by aversive odorants, thus providing a much stronger conceptual framework for the manuscript. While chronic manipulations do not happen naturally, changes in neuronal excitability and tone can persist for days. Interestingly, eating disorders are often chronic in nature. Thus, by driving the activity of neurons that are responsive to aversive odors or odors associated with spoiled food, it is plausible that we are modeling an anorexic state by constant activation of hypophagia circuits. We have since added further consideration of this matter to the Discussion section to highlight these tensions, identify unresolved questions, and to clarify different interpretations.

4) Figure 7 shows that stimulation of an anorexia circuit can lead animals to avoid places with food or food odors, which make sense, but this does not explain why the smell of food would activate an anorexia circuit (Figure 1). The authors should clarify this by rewriting the setup for Figure 7 (subsection “Acute activation of Vglut2BF neurons and their LH projections drives avoidance of food odors”), which at present seems to imply that Figure 7 is a test of the logical implications of Figure 1.

We fully acknowledge that this was a main conceptual gap in the previous version of the manuscript, we have since performed additional experimentation and provide new data to help clarify this point. Given that new data show aversive odorants can either override and/or selectively stimulate this anorexic circuit, we have significantly revised the data and description for the previous Figure 7 that is now Figure 10.

Reviewer #3:[…] 1) It would be helpful if more zoomed out confocal images showing the extent of viral labeling in the BF were shown. Are there cells outside of the BF that were labeled? Additionally, zoomed out images of fibers in LH would be helpful in order for the reader assess how specific this projection is for targeting the LH. Where other terminal fields (aside from the LH) discovered following labeling of Vglut2 BF neurons?

We appreciate this concern, and now provide zoomed-out confocal images of both the viral labeling and tracing. Additionally, we also include data showing the other terminal fields we identified to be targeted by Vglut2-BF neurons following presynaptic marker labeling in a revised Figure 7—figure supplement 1.

2) Are the data in Figure 1E trial averaged or example data from each neuron in only one trial?

For clarification to the reviewer, the data in the original Figure 1E was example data from each neuron in only one trial. Since Figure 1 has been significantly revised, this point has been further clarified in the text to avoid confusion.

3) Data in Figure 5C and D appear like they should be combined and analyzed with a two-way ANOVA. Same issue with Figure 6B.

We thank the reviewer for this suggestion. The data in the original figure 5C/D and Figure 6B have been combined, and a two-way ANOVA was used to analyze the grouped data. This information is now provided in the legend, and is now included within revised Figure 8 and Figure 9.

4) For microendoscopic imaging experiments, please report the optical specs for the GRIN lenses used. Including NA, focal distance, and GRIN lens length. Additionally, please report LED power parameters in mW.

We thank the reviewer for pointing out this unintentional omission of important methodological information. We have now added this information to the methods sections, and reported LED power parameters in mW.

5) How was food interaction quantified? This should be clarified.

For clarification to the reviewer, food interaction was quantified as time mice entered a flat petri dish containing food pellets. Any instance the mouse was in that area, smelling, or touching the pellets was considered interaction. At any point when the mouse began to consume the pellet, it was considered “food eating”. Further description and clarification of these descriptors and parameters have now been added to the text.

6) Discussion section, please clarify what particular gross hypothalamic lesions reduce feeding as some hypothalamic lesions (VMH lesions) can increase feeding.

We thank the reviewer for bringing up this point. The text has been appropriately changed and clarified in the text, as lesions to the lateral hypothalamus cause anorexia.